# Climate econometric models indicate solar geoengineering would reduce inter-country income inequality

Anthony R. Harding [1,2], Katharine Ricke [1,3]*, Daniel Heyen[4], Douglas G. MacMartin[5] & Juan Moreno-Cruz [6]

Exploring heterogeneity in the economic impacts of solar geoengineering is a fundamental step towards understanding the risk tradeoff associated with a geoengineering option. To evaluate impacts of solar geoengineering and greenhouse gas-driven climate change on equal terms, we apply macroeconomic impact models that have been widely applied to climate change impacts assessment. Combining historical evidence with climate simulations of mean annual temperature and precipitation, we project socio-economic outcomes under high anthropogenic emissions for stylized climate scenarios in which global temperatures are stabilized or over-cooled by blocking solar radiation. We find impacts of climate changes on global GDP-per-capita by the end of the century are temperature-driven, highly dispersed, and model dependent. Across all model specifications, however, income inequality between countries is lower with solar geoengineering. Consistent reduction in inter-country inequality can inform discussions of the distribution of impacts of solar geoengineering, a topic of concern in geoengineering ethics and governance debates.

[1] School of Global Policy and Strategy, University of California, San Diego, USA. [2] School of Economics, Georgia Institute of Technology, Atlanta, USA. [3] Scripps Institution of Oceanography, University of California, San Diego, USA. [4] Center of Economic Research, ETH Zürich, Zürich, Switzerland. [5] Mechanical and Aerospace Engineering, Cornell University, Ithaca, USA. [6] School of Environment, Enterprise and Development, University of Waterloo, Waterloo, Canada. *email: kricke@ucsd.edu

Climate change poses many risks to society and natural ecosystems, and action will be required to reduce its harms[1]. While the most straightforward and certain way to reduce these harms is by reducing, and eventually reversing, emissions of greenhouse gases, such mitigation is expensive and subject to free-rider incentives. The consequent inaction has led to consideration of intentional intervention in the climate system through solar geoengineering[2], but many are reluctant to pursue one global climate intervention to correct for another[3,4]. It is of paramount importance to understand, to the best of our abilities, the relative global and distributional socio-economic impacts of all climate change options.

Solar geoengineering is the intentional reflection of solar radiation to reduce the temperature effects of climate change. Until recently, understanding of the consequences of blocking sunlight to cool the planet was limited in comparison to our understanding of the effects of rising greenhouse gases. A decade of research has greatly increased our knowledge of what the climate effects of solar geoengineering might look like[5,6], but solar geoengineering impacts assessment still lags behind evaluations of other types of climate change[7]. This is for two reasons: first, the field is still relatively immature, and hence the type of physical climate modeling results required to drive impacts models did not exist until recently[8]. Second, the broader field of climate change impacts assessment has evolved in a way that does not easily accommodate application to solar geoengineering. For the sake of setting straightforward but meaningful climate policy targets, global or regional temperature anomalies are often used as proxies for the level of impact or damage[9], but with solar geoengineering, the correlations between temperature and other impact-relevant variables such as precipitation and ocean pH may differ substantially from the correlations between these variables under greenhouse gas-driven change[10]. This has made it difficult to translate projected climate effects of solar geoengineering into impacts on society using the standard frameworks used to compare, for example, a high and low carbon dioxide emissions scenario.

In this paper, we examine the global and distributional impacts of solar geoengineering on socio-economic outcomes using a state-of-the-art macroeconomic climate impacts assessment approach. This methodology, as developed by Dell et al.[11], Burke et al.[12], and Burke et al.[13], estimates the historical relationship between mean annual temperature and precipitation and country-level growth in economic production measured as gross domestic production (GDP) per capita. The empirically estimated climate-economy relationship is then applied to stylized climate scenarios constructed from projections of mean annual temperature and precipitation derived from multi-model ensembles of climate change and solar geoengineering model simulations[14–16]. We then evaluate how solar geoengineering may affect global economic growth and inter-country income inequality by comparing global and country-level economic outcomes across scenarios.

The empirically estimated climate impact models we apply use mean annual temperature and precipitation to measure the relationship between the climate and the economy, as measured by GDP. Factors such as climate variability and extremes are only captured by this model to the extent that they are related to the climate indicators used in these models. We cannot partition these effects from the aggregate effects using the empirical impacts estimation models we apply, and as such, considering the impact of these is outside the scope of our analysis. However, recent work using a high-resolution forecast-oriented model found that the type of solar geoengineering simulated in the GeoMIP simulation ensemble (which we apply here as well) mediates precipitation extremes over 99.6% of grid cells and reduces tropical cyclone intensity, not just mean climate response, supporting the assumption that there is a strong relationship between reduction of mean anomalies and mitigation of extremes[17]. Side-effects of solar geoengineering such as changes in ground-level UV[18] as well as impacts of elevated atmospheric $CO_2$ concentrations on ocean acidification[19,20] are similarly not incorporated.

Empirical economic climate impacts estimation methods are an area of active research and the extent to which projections applying such models can be reliably interpreted is a matter of some dispute in the climate change economics community[21]. We remain agnostic to this debate by applying a well-established methodology for climate change impacts estimation[12,13] to solar geoengineering in order to compare several illustrative future climate change scenarios with different levels of solar geoengineering on equal terms. We conduct a broad sensitivity analysis using competing econometric model specifications to illustrate which of our findings are contingent upon assumptions across various state-of-the-art impacts models.

The econometric models we estimate capture a mixture of linear and non-linear effects, different country trends, different climate variables, and growth and level effects. To allow for the influence of different climate variables on economic production, we estimate models with temperature and precipitation. As shown in Supplementary Table 1, temperature is the only climate variable found to be statistically significant across all the models. Economic outcomes may be delayed in their response to climate, so we estimate models with only contemporaneous climate variables as well as models that include lagged climate variables up to 5 years. Since it is unclear whether climate change impacts are on the level or growth of economic output, we estimate both types of models. Microeconomic evidence suggests the impact of temperature on outcomes follows a non-linear structure, so we estimate models both linear and non-linear in climate variables[22,23]. Finally, since countries may be following different economic trends, we estimate models using country-level trends. Results for all models can be found in the Supplementary Materials (Supplementary Table 1). In the text, we present results for the model used in the text of Burke et al.[13], but comparable results for other model specifications can be found in the Supplementary Materials.

Through this analysis, we find that the harms of warming and benefits of cooling both accrue disproportionately to warmer, poor, more populous countries. As such, climate-econometric models indicate that solar geoengineering would reduce inter-country income inequality. While the magnitudes of the economic impact of greenhouse gas-driven warming and solar geoengineering-driven cooling are highly model dependent, their influences on inter-country inequality are consistent.

## Results

**Four illustrative future climate scenarios.** To comparatively evaluate the impacts solar geoengineering with climate change impacts, we construct stylized climate scenarios from climate change and solar geoengineering projections widely used in impacts assessment. For projections of climate change without solar geoengineering, we utilize grid-cell level projections of temperature and precipitation by 2100 from the representative concentration pathway (RCP) 8.5, an emissions intensive scenario and the highest warming pathway among RCPs[24]. Temperature and precipitation responses for RCP8.5 are constructed from an ensemble mean of the climate models participating in CMIP5. Projections of grid-cell temperature and precipitation responses to solar geoengineering are constructed from climate model responses to the GeoMIP G1 experiment in which a solar

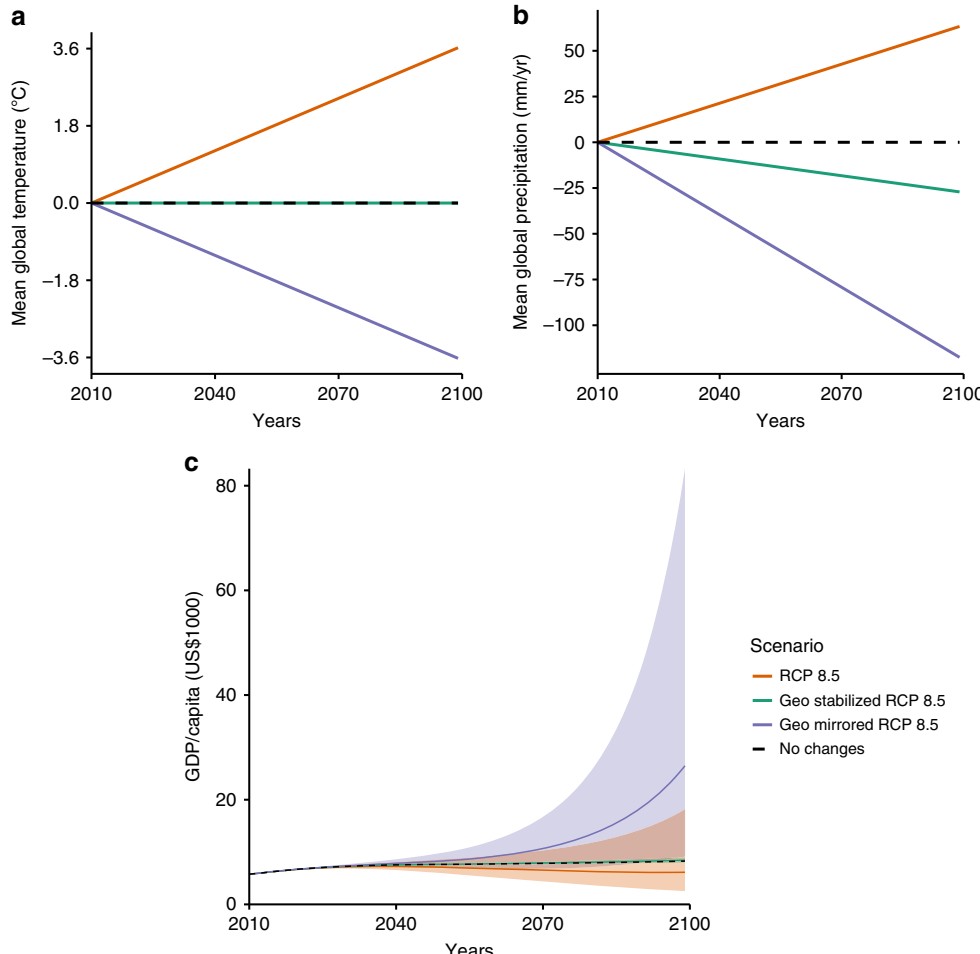

**Fig. 1 Simulated changes in climate and projected GDP/capita over the 21st century.** Curves are estimated using the model in column (1) of Supplementary Table 1 for Shared Socioeconomic Pathway (SSP) 3. **a** Change in global mean temperature and **b** change in global mean precipitation under the four illustrative climate scenarios. **c** Projected GDP/capita for the four illustrative climate scenarios where lines represent median projections and shaded area represents 95% confidence (See the "Methods" section). See Supplementary Materials for other SSPs, climate-economy model specifications (Supplementary Fig. 2).

reduction was used to offset $CO_2$ forcing[25] (See the "Methods" section). Temperature and precipitation responses for solar geoengineering are constructed from an ensemble mean of 12 climate models in the GeoMIP G1 experiment (Supplementary Table 2). We also analyze climate impacts for each of the 12 climate models individually to examine sensitivity to uncertainty in solar geoengineering climate response.

We integrate the RCP8.5 and solar geoengineering projections to simulate economic growth under four illustrative future climate scenarios (Fig. 1). These four scenarios are: no climate change, where a present-day climate is held constant, and the only simulated changes are the socioeconomic projections; RCP8.5, the highest warming scenario simulated in the CMIP5 ensemble; geoengineering-stabilized RCP8.5, in which solar geoengineering is used to stabilize global mean temperature at its present-day level despite the increased greenhouse gas concentrations associated with RCP8.5; and geoengineering-mirrored RCP8.5, a scenario in which solar geoengineering is deployed to cool the global mean temperature at the same rate of warming under RCP8.5 also despite the increased greenhouse gas concentrations associated with RCP8.5. These stylized scenarios were designed to illustrate the comparison of solar geoengineering with RCP8.5, a climate change scenario commonly utilized in climate change impact assessment.

A baseline economic growth scenario is required to apply the empirical climate impact function in projections. We use the shared socio-economic pathways (SSPs) that project key socio-economic factors such as population and economic development contingent upon challenges to adaptation and mitigation of climate change[26]. In the text, we present the results for outcomes under SSP3, the pathway associated with high challenges to both mitigation and adaptation—the conditions under which solar geoengineering seems most likely to be needed. Results for all four illustrative climate scenarios and all five SSPs can be found in the Supplementary Materials.

Changes in global temperature and precipitation for the four climate scenarios are displayed in Fig. 1a, b. The relative effects on temperature and precipitation as well as the spatial heterogeneity of impacts from solar geoengineering do not match those of anthropogenic climate change (see Supplementary Fig. 1). Solar geoengineering reduces global precipitation more per degree of cooling than $CO_2$ and other greenhouse gases increase it per degree of warming. Uniformly applied solar geoengineering also overcools equatorial regions relative to the poles.

**Marcoeconomic impacts of solar geoengineering.** When the economic impacts of solar geoengineering are estimated using the same historical evidence used to project harms from greenhouse

gas-driven warming, we observe impact model-dependent results. Following the approach of Burke et al.[12],[13], we find that solar geoengineering to stabilize global temperature mitigates the economic harms of warming-associated climate change and even provides a modest increase in global GDP (Fig. 1c). This increase is the result of the more zonally uniform global temperatures associated with canceling $CO_2$ radiative forcing with solar forcing. If anthropogenic warming is not just eliminated but solar geoengineering is used to cool the planet at a rate equal to the RCP8.5 warming rate, global GDP increases substantially due to rapid economic growth in warmer developing nations (Fig. 2b). This increase in global GDP is the result of cooling the areas of the world with high population densities that are currently warmer-than-optimal. However, these results are sensitive to econometric model specification. Supplementary Fig. 2 shows that global economic growth varies across econometric specifications as well as socioeconomic pathways.

Global results mask considerable heterogeneity in the distribution of economic losses and gains. Projections under the no-climate-change scenario and the geoengineering-stabilized RCP 8.5 scenario are similar in terms of country-level outcomes (Fig. 2c, d); no country is poorer by the end of the century than in 2010 for either scenario (Supplementary Table 3). As projected by Burke et al.[12], under RCP8.5 and SSP3, 43% of countries are poorer at the end of the century and 76% of countries are relatively poorer than they would be under SSP3 alone. Using the same impacts model, we find that under the geoengineering-mirrored RCP8.5 scenario, just 11% of countries are poorer at the end of the century and 32% of countries are relatively poorer than they would be under SSP3 alone. As shown by Supplementary Figs. 3–6 in the Supplementary Materials, the identity of countries that experience economic losses and the magnitude of their absolute or relative losses also varies across models.

**Solar geoengineering and inter-country income inequality.** From our projections we analyze differences in country-level incomes, as measured by GDP, as a metric of global income inequality. Changes in climate from climate change or solar geoengineering can additionally impact inequality across communities within the boundaries of a country. This is an important consideration for a comprehensive analysis of the impacts on inequality, however, because the models we use are identified on country-level GDP, we cannot analyze the impact on inequalities within a country's borders. The effects of each scenario on country-level economic growth, inequality, and the percentage of countries absolutely or relatively poorer varies across economic impact model specifications (see Supplementary Tables 4 and 5). However, unlike projections of global economic growth over the next century, projections of global income inequality are qualitatively consistent across models, suggesting that using solar geoengineering to negate or reverse climate change can reduce global income inequality.

Figure 3 shows the cumulative share of global GDP vs. the cumulative share of the global population (known as a Lorenz curve) in 2099 for the baseline SSP3 scenario. Absent consideration of climate change, most long-term economic projections anticipate some degree of country-level income convergence over the coming century, that is, a narrowing of the global income distribution over time. This is illustrated by the black curve. With no climate change, an end-of-century Lorenz curve is less convex than that of the present day (gray dashed line), indicating a decrease in global income inequality. These gains in equality are eliminated under RCP8.5 but are restored in a geoengineering-stabilized climate. Global cooling further increases income convergence, except in the lowest-wealth quartile. (For example,

the poorest country in 2100 under the geoengineering-mirrored climate is Mongolia with $316/capita, a decrease from $860/capita in 2010.) Supplementary Fig. 7 displays the Lorenz curves across different model specifications.

**Sensitivity analysis and robustness.** In Fig. 4 we display the percentage of countries that gain relative to no climate change and the Gini coefficients for country GDP/capita in 2099 for the different econometric models and illustrative climate scenarios under SSP3. Gini coefficients are a widely used measure of inequality, related to the curvature of the Lorenz curves in Fig. 3, where a lower Gini coefficient indicates lower inequality. Despite significantly disparate models of how climate impacts economic growth, several consistent trends emerge. RCP8.5 (orange) consistently increases inter-country inequality and the percentage of countries with poor economic growth, whereas the geo-mirrored scenario (purple) consistently decreases inequality. For all impact models, the Gini coefficient decreases with the use of solar geoengineering. The coefficient is the lowest for the Geoengineering-Mirrored RCP 8.5 scenario. Under all but one economic impacts model, the Geoengineering-Mirrored RCP 8.5 scenario decreases the percentage of countries with a GDP loss relative to RCP8.5, and under that particular model (Model 5, an income-dependent growth model with no country time trends), geoengineering has a particularly large effect on reducing inequality.

While the effects of climate change and solar geoengineering on convergence varies somewhat depending upon the socio-economic scenario and economic impact model specification, results indicate that anthropogenic warming consistently hinders or even reverses convergence, whereas solar geoengineering enhances or accelerates it. Solar geoengineering is not perfectly equitable in countering climate change in terms of key climate indicators, but it is more equitable in economic outcomes than under a no climate change scenario[27]. These results display a consistent decrease in global income inequality with solar geoengineering across economic model specification. Likewise, this result is consistent among all SSPs.

The underlying econometric models have very different assumptions that can explain both the wide range of future global production and simultaneously the consistency of solar geoengineering's impact on global income inequality across model specification. In both cases, it is the impact on economic growth in poorer countries that drive faster economic growth under some models and consistently reduce global income inequality across all models. For example, under model specifications that are quadratic in climate variables, poorer countries, which represent a large fraction of the world's population, initially have temperatures several degrees above the estimated optimal temperature. Reducing global temperatures does little to change outcomes for richer countries clustered around the temperature optimum because of relative insensitivity to marginal changes in temperature around the optimum. However, countries far from the optimum can experience large gains due to the non-linear relationship between temperature and the economy. In linear model specifications, it is a similar mechanism where initially poorer countries drive income convergence because estimates find that only poorer countries are sensitive to changes in climate. Additionally, masking $CO_2$-driven warming with solar reduction reduces the equator-to-pole temperature gradient, bringing all countries' climates slightly closer.

This analysis only captures the projected economic effects of anthropogenic warming and solar geoengineering that are associated with annual-mean temperature and precipitation, two commonly reported climate indicators which were used to calibrate the empirical impacts models applied. Changes to annual mean temperature and precipitation are closely related

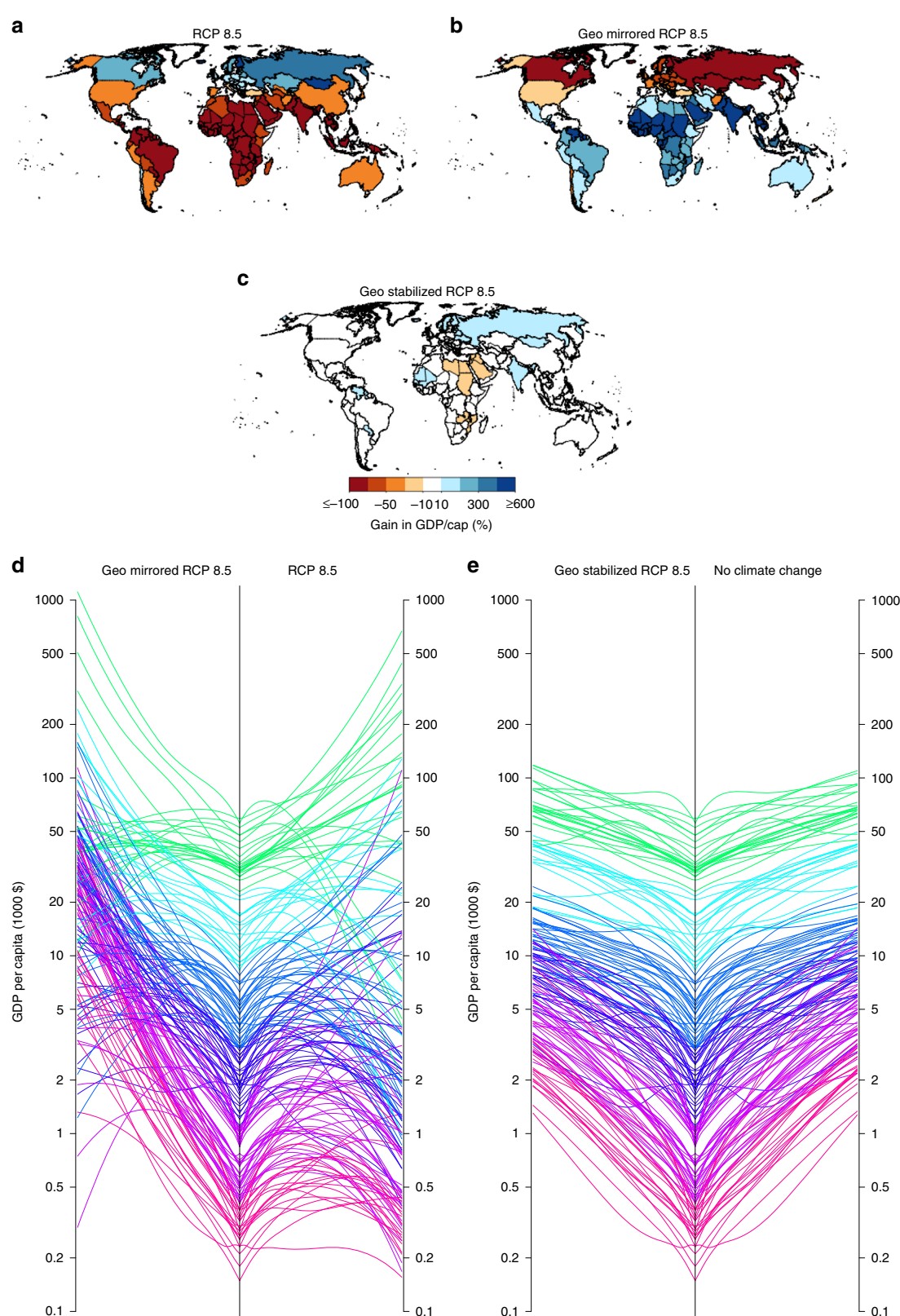

**Fig. 2 County-level income projections over the 21st century with and without solar geoengineering.** Results are estimated using the model in column (1) of Supplementary Table 1 and Supplementary Table S1 for Shared Socioeconomic Pathway (SSP_ 3. Projected percent gain in GDP per capita by 2100 relative to no climate changes for: **a** Geoengineering-mirrored RCP8.5, **b** RCP8.5, and **c** Geoengineering-stabilized RCP8.5 scenario. **d** the transient evolution of GDP per capita for each country over time under geoengineering-mirrored RCP8.5 and RCP8.5, as well as **e** the Geoengineering-stabilized RCP8.5 and SSP3 without climate change. Each line represents a specific country with color representing the country's initial GDP per capita in 2010. See Supplementary Materials for other SSPs, climate-economy model specifications (Supplementary Figs. 3 and 6).

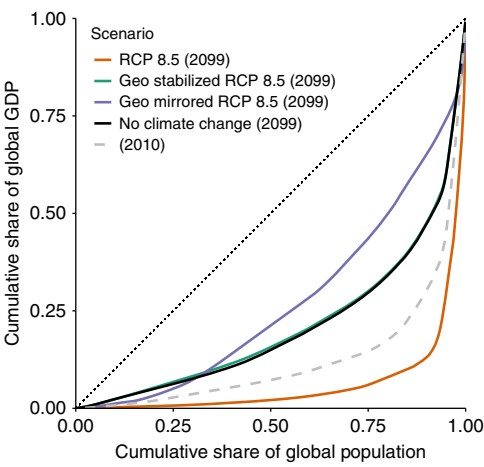

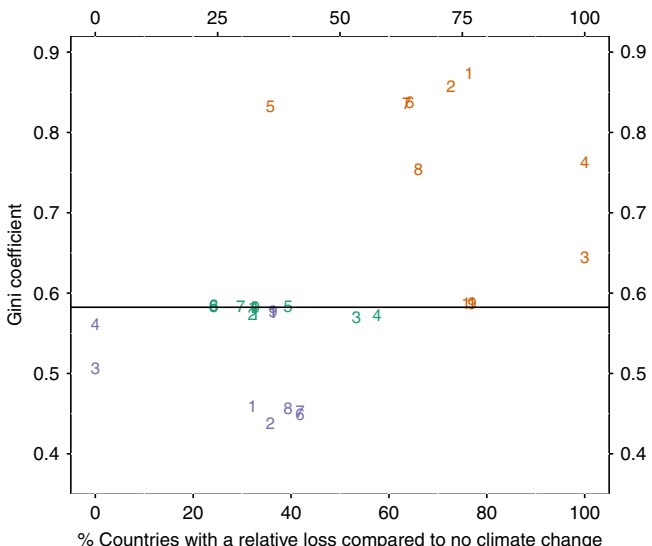

**Fig. 3 Lorenz curves of global income distribution in 2100.** Curves are estimated using the model in column (1) of Supplementary Table 1 and Supplementary Table 1 for Shared Socioeconomic Pathway (SSP) 3. Cumulative global income vs. cumulative global population, with global warming (RCP8.5, orange), no warming, geoengineering stabilized global temperature (geo-stabilized, green) and global cooling (geo-mirrored, purple). Lorenz curve for present day income distribution is indicated by dashed line. The distribution that would be observed with perfect equality is represented by the dotted line. See Supplementary Materials for other SSPs, climate-economy model specifications (Supplementary Fig. 7).

**Fig. 4 Percentage of countries with a relative loss compared to no climate change versus country-level Gini Coefficients in 2099.** Values represent median projections for Shared Socioeconomic Pathway 3 for RCP8.5 (orange), geoengineering-stabilized RCP8.5 (green) and geoengineering-mirrored RCP8.5 (purple) simulations. Numbers represent models specified as follows: Model 1 estimates a pooled growth model with quadratic temperature and precipitation, year fixed effects, and a quadratic country time trend. Model 2 estimates a growth model with quadratic temperature and precipitation and lags up to 5 years, year fixed effects, and a quadratic country time trend. Model 3 estimates a growth model with quadratic temperature and precipitation for rich and poor countries separately, year fixed effects, and a quadratic country time trend. Model 4 estimates a growth model with quadratic temperature and precipitation for rich and poor countries separately lagged up to 5 years, year fixed effects, and a quadratic country time trend. Model 5 estimates a growth with linear temperature separately for rich and poor countries, region-year fixed effects, and no country time trend. Model 6 estimates a pooled growth model with quadratic temperature, region-year fixed effects, and no country time trend. Model 7 estimates a pooled growth model with quadratic temperature and precipitation, region-year fixed effects, and no country time trend. Model 8 estimates a pooled growth model with quadratic temperature and precipitation lagged up to 5 years, region-year fixed effects, and no country time trend. Model 9 estimates a pooled levels model with quadratic temperature and precipitation, region-year fixed effects, and a quadratic country time trend. Model 10 estimates a pooled levels model with quadratic temperature and precipitation, year fixed effects, and a quadratic country time trend. Model 11 estimates a pooled levels model with quadratic temperature and precipitation, region-year fixed effects, and no country time trend.

to changes in extremes, both for GHG-driven warming[28] and solar geoengineering[17]. Impacts unaddressed by solar geoengineering, such as ocean acidification and $CO_2$ fertilization, and side-effects such as changes in ground-level UV, are potentially important factors in the economic assessment of both solar geoengineering and conventional climate change. Likewise, effects such as variability in extremes and sea level rise that may be addressed by solar geoengineering are outside of the scope of the empirical methodologies applied in this analysis. However, even a conservative interpretation of studies of the economic impacts associated with ocean acidification[19,20,29] and elevated ground-level UV[30], seem to indicate such costs would be small compared to the temperature-driven impacts of climate change.

**Uncertainty about the significance of precipitation changes.** The impacts that solar geoengineering may have on global and regional hydrological change has been a focus of considerable study and concern over the past decade[31–34]. This study and others have found limited effects of precipitation on economic growth[11,35,36], meaning our projected outcomes are mainly driven by temperature. Both greenhouse gas-driven warming and solar geoengineering are expected to decouple the historical regional relationships between temperature and precipitation in a way that is not necessarily well-accommodated by empirical impacts models. While historically, annual precipitation and temperature are negatively correlated most areas over land (Fig. 5a), the sign of the projected relationship between precipitation and temperature changes for nearly half of all countries in the analysis (Fig. 5b, c). Lack of cross-sectional variation in correlations could prove problematic when projections are then made using a model that includes country fixed effects[37,38] in which the value of a base climate state are aggregated with the value of non-physical properties such as economic and political institutions.

To examine the impacts of uncertainty about precipitation responses to solar geoengineering on economic outcomes, we apply the 12 individual GeoMIP climate ensemble members

(Supplementary Table 2) to project GDP per capita for each climate model individually. Climate variable output from individual model ensemble members span a broader range of temperature and precipitation responses, which translates into greater uncertainty in global economic impacts (Supplementary Fig. 8. However, across projections for each of the climate models, our finding that solar geoengineering reduces global income inequality still holds (Supplementary Table 5). Further, when we apply the ensemble mean temperature response and only vary precipitation response across solar geoengineering climate models to analyze sensitivity to uncertainty in the hydrological impact of solar geoengineering, we find little variation in economic impacts for the different models (Supplementary Fig. 9). This suggests that, counter to common conceptions about solar geoengineering impacts, uncertainty about temperature responses is a more

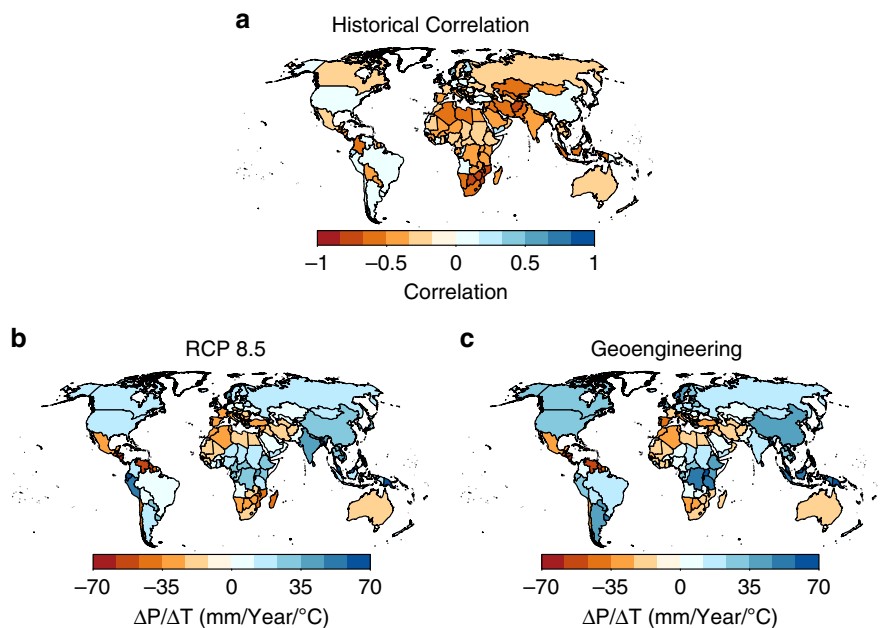

**Fig. 5 Historical and projected relationship between surface temperature and annual precipitation. a** Historical correlation between temperature and precipitation. Change in precipitation relative to change in temperature projected by (**b**) a CMIP5 ensemble for RCP8.5 and by (**c**) a GeoMIP ensemble for solar geoengineering to reduce the global the mean temperature by an equal amount as the warming under RCP8.5. The sign of the projected relationship between precipitation and temperature changes for 76 of 165 countries under RCP8.5 and 73 of 165 countries under solar geoengineering.

important driver of uncertainty about economic impacts than uncertainty about precipitation responses.

## Discussion

Our findings indicate a potentially large global economic gain from solar geoengineering, if implemented. This does not necessarily indicate that a globally governed deployment strategy would resemble our stylized scenarios. Heterogeneous impacts suggest that the scenario with greatest global economic growth may not be politically feasible under a globally governed system. Furthermore, the scenario with the largest global economic gains is associated with relative losses for the lowest wealth quartile (Fig. 3). Using the methodology employed in this analysis to evaluate potential solar geoengineering by different governance structures, or lack thereof, are important topics for future research but beyond the scope of this paper.

For purposes of this analysis we generated stylized geoengineering scenarios based on those that have been widely used by climate modelers because our interest is to explore how extreme geoengineering might affect economic growth and inequality. Among the many additional important questions that are beyond the scope of the analysis is how the exact kinds of geoengineering interventions might affect these same outcomes. Already in the broader literature, some scholars have imagined ideal global geoengineering schemes while others see geoengineering emerging in more haphazard ways—initially with actions by governments that may act unilaterally and then, later, with a wider group that sees systemic responses as better than uncoordinated unilateral actions[39–42]. Understanding whether and how these different kinds of deployment scenarios impact outcomes an important topic for future research[43].

Finally, these conclusions are dependent on the historically trained climate-econometric models being valid in predicting future impacts of geoengineering, but if these models are not valid for geoengineering, we should also expect them to also be invalid for GHG-driven climate change. As macroeconomic analyses have become a standard tool for climate change impact[11–13,23], it is essential to apply these same tools to evaluate the impacts of solar geoengineering in order to evaluate policy alternatives on equal footing. If our application and results induce skepticism, this may indicate that the empirical macroeconomic impacts assessment approach is inappropriate to apply in projecting future climate damages in general, whether solar geoengineering is a component of that future change or not. If this modeling approach accurately identifies the climate-economy relationship independent of the driving cause of climate variation, then empirical macroeconomic impacts models suggest that, depending on how it is ultimately deployed, if ever, solar geoengineering could potentially ameliorate some of the projected economic impacts of warming. There is no apparent reason that this empirical modeling approach and resulting climate change impact projections would be appropriate to apply in one instance and not the other.

Our results are not consistent with several prevailing concerns about the potential impacts of climate geoengineering: that solar geoengineering favors developed countries over developing ones, that it would have large residual economic impacts, or that maintaining a climate close to present day is clearly preferable[44]. There are substantial uncertainties associated with the models applied in this study, but the reduction of inter-country inequality is consistent across all socioeconomic scenario, climate model and economic model combinations. The insignificance of precipitation that is suggested by empirical impacts models results renders large hydrological changes associated with solar geoengineering unimportant even if intuitively this appears to be a highly consequential side effect. These inconsistencies between solar geoengineering impact assessment and state-of-the-art climate econometrics need to be addressed and resolved in order to provide a sound basis for climate risk mitigation decision-making.

We have presented results based on stylized scenarios that are unlikely to be politically or legally feasible. However, the strategic incentives implied by the results highlights the need for further work on the global governance of solar geoengineering. Following the extensive body of literature on solar geoengineering

governance[45], our findings underscore that a robust system of global governance will be necessary to ensure that any future decisions about solar geoengineering deployment are made for collective benefit.

## Methods

**Climate projections.** The projections of anthropogenic climate change are an ensemble mean of the change in precipitation and near-surface temperature in 2081–2100 relative to 1986–2005 from all global climate models participating in CMIP5 (Supplementary Figs. 1a, b). The grid-cell level climate projections are aggregated to the country-level population-weighted means by using the grid-cell level distribution of the global population in 2000 (Supplementary Fig. 1e). We interpolate annual climate change for RCP 8.5 under the assumption that temperature and precipitation follow a constant linear trend from 2010 through 2100[12]. This is consistent with temperature and precipitation trends under RCP 8.5.

The projections of changes in temperature and precipitation from solar geoengineering are constructed from the ensemble mean of 12 models contributing to GeoMIP (Supplementary Table S2)[25,46,47]. These projections represent the respective change in each climate indicator for a degree Celsius decrease in global temperature from solar geoengineering (Supplementary Figs. 1c, d; note that the shift in equator-to-pole temperature gradient may be different for different solar geoengineering strategies). Solar geoengineering projections are aggregated to country-level population-weighted means using the population distribution in 2000[12]. In our illustrative scenarios, we consider two levels of solar geoengineering. The first, Geoengineering-Stabilized RCP 8.5, deploys solar geoengineering to counter increases in the global mean temperature from RCP 8.5 to stabilize the global mean temperature at 2010 levels throughout the 21st century. The second, Geoengineering-Mirrored RCP 8.5, deploys solar geoengineering to decrease the global mean temperature at the same rate it would increase under RCP 8.5 without any solar geoengineering. These two scenarios are illustrated in Fig. 1a, b.

**Economic impact function.** Our impact function estimations start with direct replications of Dell et al.[11], Burke et al.[12,13]. For the econometric estimation of the historical climate-economy relationship, we follow the approach of Burke et al.[12]. Using historical data on interannual and inter-country variation in annual average temperature and precipitation from 1960–2010 for 165 countries[48] and GDP per capita[49], they estimate the historical non-linear relationship between key climate indicators and growth in GDP per capita. (See Supplementary Table 1 for regression results.)

**Economic projections.** For the economic projections, we follow the approach of Burke et al.[12] with a small extension. The economic projection consists of three steps. The first step is to select one of the five SSPs. This choice of an SSP determines, for each of the 165 countries, a baseline projection of population and per capita GDP growth for each year between 2010 and 2099[26]. This baseline projection implicitly assumes that climate indicators do not change over the course of the century and therefore represents the growth profile in the no-climate-change scenario. The second step (for the remaining three scenarios that feature a change in climate conditions) is to iteratively adjust, for each country separately, the growth projection according to changes in climate indicators. The basis for this adjustment is the impact function (see Economic Impact Function above) that describes the historical climate-economy relationship. For a given year, the growth rate is modified upwards or downwards according to a country's position on the climate impact function in that year relative to their climate in 2010. In this way, we obtain a growth profile over time for each country. Finally, the third step of the economic projection is to apply these annual growth rates to the initial GDP/capita of each country in 2010 to evaluate each country's GDP/capita throughout the century.

**Uncertainty analysis.** To test consistency of our findings across specifications of the climate-economy relationship we estimate multiple impact models. While in the main text we follow the model used in the text of Burke et al.[12], the Supplementary Materials show results for a variety of alternative specifications. While the specification used in the text follows the assumption that growth rates only depend on present climate conditions, we also estimate models where economic impacts depend on climate conditions in the previous five years (lagged). In addition to using uniform impact function (pooled), we allow response functions to vary across countries by estimating models with a separate climate-economy relationship for rich and poor countries. While microeconomic evidence suggests a non-linear response structure to temperature, we estimate both linear and non-linear model specifications. Finally, since it is unclear whether the climate-economy relationship impacts levels or growth of economic output, we estimate both types of model.

To account for uncertainty in the estimated historical response functions, we use a bootstrap estimation of the econometric impact functions ($N = 1000$) in which countries are sampled with replacement. For median results, we use the 50% quantile projections. To describe 95% confidence intervals, we use 2.5% and 97.5% quantile results.

## Data availability

All data generated and used in this analysis can be accessed at https://github.com/klricke/solar-geoengineering-econometrics-inequality.

## Code availability

All code generated and used in this analysis can be accessed at https://github.com/klricke/solar-geoengineering-econometrics-inequality.

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

## Acknowledgements

The authors thank Sikina Jinnah and David Victor for helpful input on the global governance implications of this paper. K.R. and A.H. thank the Deep Decarbonization Initiative for support of this research. J.M.C. acknowledges support from the Canada Research Chairs initiative.

## Author contributions

K.R., D.H. and J.M.C. conceived of the idea for the study. D.G.M. provided the geoengineering simulation emulator. A.H. and K.R. designed the simulations and analysis. A.H. performed the analysis. A.H. and K.R. wrote the manuscript. All authors discussed the results and provided input on the manuscript.

## Competing interests

The authors declare no competing interests.
