## [Peer Review File · Nature Communications]

Reviewers' comments:

Reviewer #1 (Remarks to the Author):

Review of Harding, Nature Communications

Summary: the authors argue that SRM reduced global inequality compared to a high emission scenario without.

Recommendation: I am a climate modelling person and have limited insight into economic models, so other need to comment in more detail on that part.

I have mixed feelings about this paper. The results appear to be technically correct, and analysis framework makes sense, and model uncertainties are discussed to the extent possible. The writing is clear, and the topic is no doubt a relevant one. My main concern is how much of what is done here applies to the real world, and can or should be interpreted. Many things are known to be highly simplified in economic models, and certainly using historic relationships is questionable (the authors even discuss that). I do not have an alternative to offer, but the assumptions are so questionable that the real issue is whether it would be more honest as scientists to say that this is playing around with toy models, and to say we don't know whether that has much relevance for the real world. The same arguments of course apply to many other similar studies.

Apart from the assumptions in economic models, the study for example only looks at annual mean changes, even though in some regions winters get wetter and summers get drier. It ignores changes in variability, in extremes, sea level, UV, etc. When costs are strongly affected by extreme events, and we know changes in solar affect the water cycle differently from greenhouse gases, how can we justify making such assumptions? The authors even mention some of these points e.g. in line 245ff, saying that the models may not be appropriate.

Reading those paragraphs I would conclude that basically nothing can be concluded because of the large uncertainties and questionable assumptions. Yet the abstract simply mentions further research, and prominently discusses "robust" results. Even more surprising is that the title states the result as a fact, no "may" or "model indicates". In my view, the results here are interesting, and they provide insight into economic models or regional distributions of damages, but the title and abstract do not reflect the limited confidence that the authors themselves appear to have.

I predict that the conclusions are going to be misrepresented in the media. I would never object to the publication of such results, this is an important topic and research in this field is needed, but given the politically charged debate about geoengineering I believe we are not doing anyone a favour by pretending to have a robust result if we don't, whether for or against geoengineering. I cannot argue that the results here are wrong in a technical sense, or that the modelling has not been done carefully, but in my view the uncertainties are so large, and the simplifications are so big, that I personally would not dare to draw any conclusions about the real world. In that sense I would not recommend publication, but I would encourage the authors to submit a revised manuscript to a technical journal, with a more careful discussion of uncertainties, and a very different framing.

There appears to be a strong but untested assumption on pattern scaling, which may be robust for some quantities but maybe not for others (doi: 10.1002/2013JD021258)

Line 58: use -> used?

Reviewer #2 (Remarks to the Author):

The paper is a very interesting one, and examines the much debated issue of solar geoengineering from an economic and equity perspective. This is a novel idea which makes the research up for a useful contribution to the debate.

The research has one big concluding message- solar geoengineering leads to improvement in global economic equity, as compared to a high global warming and climate impact (and no solar geoengineering) scenario. This is a strong conclusion, and if proven robustly, provides a fillip to the argument of employing solar geoengineering technologies. It needs to be hence ensured that the conclusion is robust, and the analytical approach as well as the assumptions are robust. I see some major limitations on both fronts, as given below:

1. Analytical approach limitation

The authors mention that they estimate the impact of change in mean temperature and mean precipitation (in BAU, as well as solar geoengineering scenarios) on the economy. This is the Achilles heel of their model. Various IPCC assessments across years have shown that it is not just the means, but the variations across time and space of both temperature and precipitation, that really matter for the sectors that are most vulnerable to variations in these, e.g. the agricultural sector. The economic model that the authors have used is a published and peer reviewed model, and is based on mean values of these variables. This approach could be defensible only if one is interested in the average impact of these variables on the GDP. For an issue related to equity, however, the model has to better represent the variance along with means, and how variance/uncertainty leads to inequitable outcomes. Doing a high level analysis GDP and growth doesn't suffice. The used model could at best show that maybe equity between countries represented by high level GDP and growth could increase or decrease. The approach glosses over the real equity issue of what happens to climate dependent communities within the boundaries of a country.

2. Input/Assumption related to impact of solar geoengineering on precipitation patterns.

It is intuitively clear that solar geoengineering will lead to reduction in average temperatures as well as have a positive impact on reducing the negative uncertainties related to temperature variation, and hence positive outcome for the world. By reducing climate change impacts, it will also lead to reduction in inequity as climate change is expected to impact the poor most. This, however, is not the case with precipitation. There is not conclusive evidence in the literature on the impacts of solar geoengineering on global hydrological cycle, and precipitation patterns across different regions and countries. Some of the research shows that depending on where sulphate particles are injected, solar geoengineering could play havoc with the monsoon patterns in the tropics. When climate impacts research itself finding it challenging to better understand the impacts of climate change on the hydrological cycle, it is not hard to understand the challenges associated with understanding the impact of solar geoengineering. The authors try to get over this issue by taking results of solar geoengineering's impact on temperature and precipitation from dedicated scenario runs, and using this information in their economic model. This, however, cannot be an acceptable approach when there is a lack of published peer review literature and a scientific agreement on this issue of impact of solar geoengineering on the hydrological cycle, be it mean, variances, and intensity. Having a robust understanding of this issue is critical, because people dependent on agriculture are most often economically weaker section of the society, be it in the developed or the developing world. Negative impacts on precipitation patterns due to solar geoengineering interventions will only increase the inequity between and within regions, depending on how the impacts due to solar geoengineering are distributed across the world. If the authors are able to cite a reasonable number of peer reviewed research papers on solar geoengineering impacts on temperature and precipitation (which I don't think would be possible at least at this stage) and if these papers have agreement on some main outcomes, then this would be reliable information to be used in the economic model.

To conclude, the issue that the authors have tried to address is a very interesting issue, and if analysed robustly can lead to moving the debate in a particular direction, which is why one has to be very sure about the analytical approach and assumptions. The analytical approach appears to view the equity issue from a country level perspective, while the equity debate is a much more deeper debate and goes inside country boundaries. The scientific literature and understanding of solar geoengineering's impact on precipitation is at very nascent stage. Given these limitations, the paper is not fit for publication as of now. It is entirely plausible. however, that in the future if the underlying information on the impact of solar geoengineering on precipitation patterns becomes robust then this information can be used in economic models given that they treat the equity issue in a deeper way.

Paper NCOMMS-18-38238-T

Title: Climate econometric models indicate solar geoengineering may reduce global income inequality

Dear Editor(s) and Reviewers,

We would like to thank both of the reviewers for their comments and suggestions. They raise important concerns about the assumptions made and uncertainties involved when applying empirical econometric climate impact assessments to project future outcomes. In this paper, our goal is to apply a widely-deployed methodology for evaluating greenhouse gas-driven climate change to solar geoengineering in order to compare futures with and without solar geoengineering on equal footing. There is still significant debate about the correct statistical and economic assumptions when modeling climate impacts using empirical methodologies. To account for this, we apply multiple models covering not just those used by the most influential papers in this field¹⁻³, but additional model specifications from a recent working paper⁴. In fact, we find that the global GDP impacts are highly model dependent (for both GHG-driven change and solar geoengineering). However, our finding of increased convergence in income inequality between countries increases with the application of solar geoengineering across all models.

While the assumptions made when applying this methodology are strong, we illustrate in our paper that there is no reason to believe the assumptions that are valid for application in climate change impacts assessment would be too strong for our application to solar geoengineering. Whatever shortcomings in empirical econometric climate impacts projections would also hold here, but they would not be specific to their application to solar geoengineering.

Both reviewers' feedback raise two primary issues, uncertainties relating to the impact of solar geoengineering on precipitation and the role of changes in climate variability and extremes. We address both of these important comments in detail below in response to the reviewers' feedback. To illustrate our argument, we have added to our discussion of methodological assumptions and their impact on the conclusions we can draw, we have changed the title and language in our paper to better reflect concerns about misinterpretation of our findings, and we have added additional analysis across solar geoengineering climate models to show our findings are consistent across uncertainties in the impact of solar geoengineering on the hydrological cycle.

We believe the revised manuscript is now stronger and more comprehensive to meet the standards of *Nature Communications*. Thank you again for all of your comments. Detailed responses to the referees' feedback can be found below. Reviewers' comments can be found in **bold** with our responses to each comment below them. All line numbers listed refer to the revised manuscript.

Sincerely,
Anthony Harding

Reviewer #1

Summary: the authors argue that SRM reduced global inequality compared to a high emission scenario without.

I have mixed feelings about this paper. The results appear to be technically correct, and analysis framework makes sense, and model uncertainties are discussed to the extent possible. The writing is clear, and the topic is no doubt a relevant one.

We appreciate these positive comments.

My main concern is how much of what is done here applies to the real world, and can or should be interpreted. Many things are known to be highly simplified in economic models, and certainly using historic relationships is questionable (the authors even discuss that). I do not have an alternative to offer, but the assumptions are so questionable that the real issue is whether it would be more honest as scientists to say that this is playing around with toy models, and to say we don't know whether that has much relevance for the real world. The same arguments of course apply to many other similar studies."

In this paper, our intent is to take an established methodology that has been used to project impacts of global warming (most recently in *Nature* in 2018³) and see what this impacts estimation methodology implies about solar geoengineering. As we write in our conclusions, the validity of our findings is contingent upon the validity of the methodology, i.e.:

"If our application and results induce skepticism, this may indicate that the empirical macroeconomic impacts assessment approach is inappropriate to apply in projecting future climate damages in general, whether solar geoengineering is a component of that future change or not. If this modeling approach accurately identifies the climate-economy relationship independent of the driving cause of climate variation, then empirical macroeconomic impacts models indicate that solar geoengineering is a tool that could be applied to grow the global economy and reduce global income inequality. There is no scientific reason that this empirical modeling approach and resulting climate change impact projections would be appropriate to apply in one instance and not the other."

The methodology we apply was used to project the benefits of meeting a 1.5 degree temperature target using emissions reductions in a paper published in 2018 in *Nature*. There is a precedent for believing this is a rigorous and state-of-the-art impacts estimation methodology. Our contention is that there is no scientific reason to presume it is valid for climate impact assessments, such as Burke et al. (2018), but not assessments of solar geoengineering.

In defense of the empirical macroeconomic impacts estimation methodology developed by Burke et al (2015)², the simplicity of the models applied is intentional. Burke et al. emphasize the utility of their approach for capturing both observed and unobserved influences of climate on economic growth. Section B in the appendix of Burke et al. (2015) contains a detailed discussion [<https://media.nature.com/original/nature-assets/nature/journal/v527/n7577/extref/nature15725-s1.pdf>].

We briefly summarize the debate surrounding the use of historical relationships on lines 86-93 where we added the following text.

“Empirical economic climate impacts estimation methods are an area of active research and the extent to which projections applying such models can be reliably interpreted is a matter of some dispute in the climate change economic community⁴. We remain agnostic to this debate by applying a well-established methodology for climate change impacts estimation^{2,3} to solar geoengineering in order to compare several illustrative future climate change scenarios with different levels of solar geoengineering on equal terms.”

To ensure our intent is clear from the onset, however, we have revised our title, abstract and introduction in order to make this framing more explicit prior to presenting our results.

Apart from the assumptions in economic models, the study for example only looks at annual mean changes, even though in some regions winters get wetter and summers get drier. It ignores changes in variability, in extremes, sea level, UV, etc. When costs are strongly affected by extreme events, and we know changes in solar affect the water cycle differently from greenhouse gases, how can we justify making such assumptions? The authors even mention some of these points e.g. in line 245ff, saying that the models may not be appropriate.”

We agree that the analysis of changes in variability and extremes is important for a comprehensive assessment of Solar Geoengineering. But such analysis goes beyond the scope of this paper. Here, we follow the well-established and influential climate impact literature with the aim of understanding the implications for solar geoengineering. The econometric model applied is calibrated on annual mean changes and thus cannot capture changes in variability, extreme events, and other such factors. Note that while these models are identified on annual means, to the extent that there is a relationship between mean changes and extremes, as the literature suggests there is⁵, this model will indirectly capture the impact of these changes. Additionally, recent work finds that the application of solar geoengineering would not worsen the intensity of tropical storms⁶.

As the referee notes, we discuss these points and their implications for our conclusions. We have adjusted the discussion of the limitations of the methodology on lines 75-84 and 240-252 to discuss that variability in extremes, changes in sea level, and other such factors are outside the scope of our analysis because they are beyond the scope of the methodology we follow.

Reading those paragraphs I would conclude that basically nothing can be concluded because of the large uncertainties and questionable assumptions. Yet the abstract simply mentions further research, and prominently discusses “robust” results. Even more surprising is that the title states the result as a fact, no “may” or “model indicates”. In my view, the results here are interesting, and they provide insight into economic models or regional distributions of damages, but the title and abstract do not reflect the limited confidence that the authors themselves appear to have.

After considering these comments, we agree that both the title and the use of the word “robust” could be misleading. Within the econometric climate impacts methodology, we evaluate impacts for uncertainties across and within underlying econometric model specification, as well as across

potential socio-economic and climate trajectories. Within this methodology, our finding that solar geoengineering reduces global income inequality holds across all uncertainties. So from a strictly statistical standpoint, the results are “robust.” However, as discussed above, there are a number of reasons why empirical macroeconomic climate impact estimation methodologies may be imperfect or incomplete as applied to projections of future change. We have accordingly adjusted the title of the manuscript and changed our language around the robustness of our results. Specifically, we have changed the title of the manuscript to “*Climate econometric models indicate solar geoengineering may reduce global income inequality*”. Additionally, we have replaced the use of “*robust*” throughout the paper with “*consistent across model specifications*”. We believe this is more representative of our results being consistent across models and uncertainties in this state-of-the-art econometric methodology.

I predict that the conclusions are going to be misrepresented in the media. I would never object to the publication of such results, this is an important topic and research in this field is needed, but given the politically charged debate about geoengineering I believe we are not doing anyone a favour by pretending to have a robust result if we don't, whether for or against geoengineering. I cannot argue that the results here are wrong in a technical sense, or that the modelling has not be done carefully, but in my view the uncertainties are so large, and the simplifications are so big, that I personally would not dare to draw any conclusions about the real world. In that sense I would not recommend publication, but I would encourage the authors to submit a revised manuscript to a technical journal, with a more careful discussion of uncertainties, and a very different framing.

We agree with the referee on two points here: first, that political considerations should not preclude publication of scientifically sound and novel results, and second, that these results could easily be misinterpreted by the media. On the second point, as noted above, we have changed the title of the manuscript to “*Climate econometric models indicate solar geoengineering may reduce global income inequality*”, revised our language around the robustness of our results, and more thoroughly discussed the assumptions made by this methodology and their justification.

There appears to be a strong but untested assumption on pattern scaling, which may be robust for some quantities but maybe not for others (doi: 10.1002/2013JD021258)”

It is correct that pattern scaling will not hold for all climate variables. However, for the variables we use in our analysis, annual mean temperature and precipitation, this assumption has been well tested. For details, see the following reference upon which our country-level temperature and precipitation projections are based:

⁷MacMartin DG, Kravitz B (2016) Dynamic climate emulators for solar geoengineering. *Atmospheric Chem Phys* 16(24):15789–15799.

Line 58: use -> used?

Thanks, this typo has been fixed.

Reviewer #2

The paper is a very interesting one, and examines the much debated issue of solar geoengineering from an economic and equity perspective. This is a novel idea which makes the research up for a useful contribution to the debate.

We thank the referee for these encouraging comments.

The research has one big concluding message- solar geoengineering leads to improvement in global economic equity, as compared to a high global warming and climate impact (and no solar geoengineering) scenario. This is a strong conclusion, and if proven robustly, provides a fillip to the argument of employing solar geoengineering technologies. It needs to be hence ensured that the conclusion is robust, and the analytical approach as well as the assumptions are robust. I see some major limitations on both fronts, as given below:

1. Analytical approach limitation

The authors mention that they estimate the impact of change in mean temperature and mean precipitation (in BAU, as well as solar geoengineering scenarios) on the economy. This is the Achilles heel of their model. Various IPCC assessments across years have shown that it is not just the means, but the variations across time and space of both temperature and precipitation that really matter for the sectors that are most vulnerable to variations in these, e.g. the agricultural sector. The economic model that the authors have used is a published and peer reviewed model, and is based on mean values of these variables. This approach could be defensible only if one is interested in the average impact of these variables on the GDP.” For an issue related to equity, however, the model has to better represent the variance along with means, and how variance/uncertainty leads to inequitable outcomes. Doing a high level analysis GDP and growth doesn't suffice. The used model could at best show that maybe equity between countries represented by high level GDP and growth could increase or decrease. The approach glosses over the real equity issue of what happens to climate dependent communities within the boundaries of a country.

The impacts models applied in our analysis are identified on variation in mean annual climate. (In Burke et al. (2015), models are identified on inter-annual variability in absolute mean temperature and precipitation both within and across countries. In Dell et al. (2012), the models are identified on relative temperature and precipitation changes.) As detailed in our response to Referee 1 above, while these impacts models are identified on annual means, they will still capture variability and extremes if there is a correlation between impacts indicators and variability and extremes. To clarify, we added the following text on lines 75-84:

“The empirically estimated climate impact models we apply use mean annual temperature and precipitation to measure the relationship between the climate and the economy, as measured by GDP. Factors such as climate variability and extremes are only captured by this model to the extent that they are related to the climate indicators used in these models. We cannot partition these effects from the aggregate effects using the empirical impacts estimation models we apply, and as such, considering the impact of these this is outside the scope of our analysis. However, recent work demonstrates that solar geoengineering generally mediates climate extremes, and rarely exacerbates them⁶.”

As we emphasize in our conclusions, the issues addressed by this comment are applicable for the general empirical macroeconomic methodology we apply, a prominent peer-reviewed approach to estimating impacts of climate change (not just our application to solar geoengineering).

As the reviewer notes, using population-weighted mean temperature and precipitation, we are able to examine impacts on global and country-level incomes. Because we are interested in cross-country inequality, our approach is meant to capture the effect of mean temperature and precipitation on GDP, exactly as suggested by the referee. Moreover, aggregate income or GDP is a commonly used measure when discussing in the economic development literature⁸.

The referee makes a fair point that the inequality we analyze in this paper is only one type of inequality. However, whether or not the “real equity issue” is within-country inequality is debatable. Our findings are relevant to international inequality or the convergence of country-level incomes over time, an important -- not comprehensive -- measure of global inequality. In fact, the international community concerned with global inequality focuses on across-country inequality just as much they do in within-country inequality⁸. Both forms of inequality are important and our methodology is suitable for analyzing the important issue of cross-country inequality. We have added text to clarify this point and highlight that, while outside the scope of our analysis, within-country distributional issues are also important when considering the impacts of climate change on human welfare.

“The empirical methodology and models we use are able to capture differences in country-level income, as measured by GDP, between countries as a metric of global income inequality. Changes in climate from climate change or solar geoengineering can additionally impact inequality across communities within the boundaries of a country. This is an important consideration for a comprehensive analysis of the impacts on inequality, however, because the models we use are identified on country-level GDP, we cannot analyze the impact on inequalities within a country’s borders. Moreover, we are conscious that our framework reflects presuppositions embedded in the modelling assumptions commonly applied to solar geoengineering and these presuppositions are particularly important when analyzing inequality outcomes⁹.”

2. Input/Assumption related to impact of solar geoengineering on precipitation patterns.

It is intuitively clear that solar geoengineering will lead to reduction in average temperatures as well as have a positive impact on reducing the negative uncertainties related to temperature variation, and hence positive outcome for the world. By reducing climate change impacts, it will also lead to reduction in inequity as climate change is expected to impact the poor most. This, however, is not the case with precipitation. There is not conclusive evidence in the literature on the impacts of solar geoengineering on global hydrological cycle, and precipitation patterns across different regions and countries.

The referee is correct that the impact of solar geoengineering on global and regional precipitation is more uncertain than its effect on temperature. This is not, however, a feature that is unique to projections of solar geoengineering. The analysis in our paper applies the projections from 12 different earth system models included in both the CMIP5 and GeoMIP ensembles and spans a

range of uncertain precipitation responses. MacMartin et al. (2015) illustrates that the precipitation response for climate change leads to large shifts in precipitation, which in most areas of the world are then reduced by solar geoengineering, and that there is very little model agreement for the small number of areas for which geoengineering exacerbates the precipitation effects of greenhouse gas-driven warming^{10,11}.

In order to further address the concerns of the referees about any misrepresentation of the outcomes associated with precipitation and geoengineering in our analysis, we have conducted a supplementary analysis that includes impact projections using each of the 12 GeoMIP ensemble members individually. We analyze economic impacts applying both temperature and precipitation response for each model as well as using the ensemble mean temperature, only varying the precipitation response for each climate model. This analysis covers a wide range of hydrological responses. We illustrate in Supplementary Figure S9 that impacts are sensitive variation in climate response across models. However, Supplementary Figure S10 shows that temperature is the more important factor for uncertainty in economic impacts. In Supplementary Table S3, we demonstrate that our result that solar geoengineering decreases income inequality holds across the individual GeoMIP models. This additional analysis reinforces our results, suggesting that variation in precipitation responses across models are not a driving factor of the results. We add the following text discussing this analysis.

“To examine the impacts of uncertainty about precipitation responses to solar geoengineering on economic outcomes, we apply the 12 individual GeoMIP climate ensemble members (Supplementary Table S2) to project GDP per capita for each climate model individually. Climate variable output from individual model ensemble members span a broader range of temperature and precipitation responses, which translates into greater uncertainty in global economic impacts (Supplementary Figure S9). However, across projections for each of the climate models, our finding that solar geoengineering reduces global income inequality still holds (Supplementary Table S3). Further, when we apply the ensemble mean temperature response and only vary precipitation response across solar geoengineering climate models to analyze sensitivity to uncertainty in the hydrological impact of solar geoengineering, we find little variation in economic impacts for the different models (Supplementary Figure S10). This suggests that, counter to common conceptions about solar geoengineering impacts, uncertainty about temperature responses is a more important driver of uncertainty about economic impacts than uncertainty about precipitation responses.”

Perhaps the most important point to re-emphasize is that the impacts models applied in our analysis generally find that precipitation is not a significant predictor of economic outcomes. As we strongly emphasize in our conclusions, the historical relationships between temperature and precipitation do not necessarily hold in the forced response to *either* increasing atmospheric greenhouse gas concentrations or solar geoengineering.

Again, this paper is about comparing apples to apples. If an empirical macroeconomic approach is a legitimate one for evaluating the impacts of global warming (as the recent peer-reviewed publications in *Nature* suggest it is), then there is no scientific reason it should be considered illegitimate for projecting the impacts of solar geoengineering. In lines 269-289 we explicitly lay out our concerns about the implications of our results for illustrating shortcomings in the

underlying impacts estimation methodology, independent of its application to solar geoengineering.

Some of the research shows that depending on where sulphate particles are injected, solar geoengineering could play havoc with the monsoon patterns in the tropics.

We agree that geoengineering can be implemented in different ways, such as different injection locations, and this can have different effects. Regarding the monsoon patterns, some older studies have suggested that geoengineering can have a significant effect on the monsoon patterns, however, this result is inconsistent across models¹², and since our results are conducted with 12 different models, any such effect in these simulations would be represented in our analysis.

When climate impacts research it itself finding it challenging to better understand the impacts of climate change on the hydrological cycle, it is not hard to understand the challenges associated with understanding the impact of solar geoengineering. The authors try to get over this issue by taking results of solar geoengineering's impact on temperature and precipitation from dedicated scenario runs, and using this information in their economic model. This, however, cannot be an acceptable approach when there is a lack of published peer review literature and a scientific agreement on this issue of impact of solar geoengineering on the hydrological cycle, be it mean, variances, and intensity. Having a robust understanding of this issue is critical, because people dependent on agriculture are most often economically weaker section of the society, be it in the developed or the developing world. Negative impacts on precipitation patterns due to solar geoengineering interventions will only increase the inequity between and within regions, depending on how the impacts due to solar geoengineering are distributed across the world. If the authors are able to cite a reasonable number of peer reviewed research papers on solar geoengineering impacts on temperature and precipitation (which I don't think would be possible at least at this stage) and if these papers have agreement on some main outcomes, then this would be reliable information to be used in the economic model.

We agree that a discussion about the uncertainty in the impact of solar geoengineering on the hydrological cycle should be more prevalent in our paper. However, as we illustrate in our paper, across model specifications, precipitation is consistently not found to be a significant factor in the historical climate-economy relationship. Thus, under this empirically estimated climate impacts methodology that applies the historical climate-economy relationship, precipitation continues to not be a factor in determining the impacts of either climate change or geoengineering. As we have mentioned above, we discuss the implications of this result and concerns about this result in our paper. Further, MacMartin et al. (2015) show that the variation in hydrological cycle response across models is greater with climate change alone than with a combination of climate change and geoengineering. Recent work also suggests that solar geoengineering would not exacerbate extreme precipitation (Irvine et al., 2019).

We have done additional analysis of our findings using each of the 12 GeoMIP models individually to complement the ensemble analysis. These models span a wide range of hydrological responses

to solar geoengineering. Following our analysis, we have added Supplementary Table S3 to illustrate that our result that solar geoengineering reduces global income inequality within this methodology still holds across hydrological responses.

To conclude, the issue that the authors have tried to address is a very interesting issue, and if analysed robustly can lead to moving the debate in a particular direction, which is why one has to be very sure about the analytical approach and assumptions. The analytical approach appears to view the equity issue from a country level perspective, while the equity debate is a much more deeper debate and goes inside country boundaries.”

We again thank the referee for their positive comment about the novelty and importance of the topic. We agree that the previous version of the manuscript was unacceptably imprecise about the type of inequality that our analysis addresses. We added the following text in lines 200-204 to address this:

“The macroeconomic methodology and models we use are able to capture differences in country-level income between countries as a measure of global income inequality. Changes in climate from climate change or solar geoengineering can have additional impacts on inequality through the distribution of impacts across communities within the boundaries of a country. These are an important consideration for a comprehensive analysis of the impacts on inequality, however, because the models we use are identified on country-level GDP, this paper cannot analyze the impact on the distribution of income within a country’s borders.”

Our findings regarding global income inequality or country income convergence is an important and relevant aspect of climate change impacts, but intra-country distributional issues are an important topic for future research.

The scientific literature and understanding of solar geoengineering's impact on precipitation is at very nascent stage. Given these limitations, the paper is not fit for publication as of now. It is entirely plausible. However, that in the future if the underlying information on the impact of solar geoengineering on precipitation patterns becomes robust then this information can be used in economic models given that they treat the equity issue in a deeper way.

As noted above, while we certainly agree that the impact of solar geoengineering on global and regional precipitation is more uncertain than its effect on temperature, we respectfully disagree that the understanding is still at a very nascent stage relative to our understanding of greenhouse gases effects on precipitation. The precipitation effects of solar forcings to counteract longwave forcings of greenhouse gases has been the subject of many peer-reviewed publications over the past decade. Second only to perhaps temperature, it is the variable that has been the major focus of analyses of the GeoMIP ensemble -- a model intercomparison project with participants from more than a dozen of the state-of-the-art earth system models included in the CMIP5 and CMIP6 ensembles. Here is a sampling of just a few of the most cited papers about this topic published in the recent years:

¹³ Kravitz, B. *et al.* An energetic perspective on hydrological cycle changes in the Geoengineering Model Intercomparison Project. *J. Geophys. Res. Atmospheres* **118**, (2013).

- ¹⁴ Kravitz, B. *et al.* A multi-model assessment of regional climate disparities caused by solar geoengineering. *Environ. Res. Lett.* **9**, 074013 (2014).
- ¹² Tilmes, S. *et al.* The hydrological impact of geoengineering in the Geoengineering Model Intercomparison Project (GeoMIP). *J. Geophys. Res. Atmospheres* **118**, (2013).
- ¹⁵ Jones, A. *et al.* The impact of abrupt suspension of solar radiation management (termination effect) in experiment G2 of the Geoengineering Model Intercomparison Project (GeoMIP). *J. Geophys. Res. Atmospheres* **118**, 9743–9752 (2013).
- ¹⁰ MacMartin, D. G., Kravitz, B. & Rasch, P. J. On solar geoengineering and climate uncertainty. *Geophys. Res. Lett.* **42**, 7156–7161 (2015).
- ¹⁶ Irvine, P. J. *et al.* Key factors governing uncertainty in the response to sunshade geoengineering from a comparison of the GeoMIP ensemble and a perturbed parameter ensemble. *J. Geophys. Res. Atmospheres* **119**, 7946–7962 (2014).
- ⁶ Irvine, P. *et al.* Halving warming with idealized solar geoengineering moderates key climate hazards. *Nat. Clim. Change* **1** (2019). doi:10.1038/s41558-019-0398-8
- ¹⁷ Xia, L. *et al.* Solar radiation management impacts on agriculture in China: A case study in the Geoengineering Model Intercomparison Project (GeoMIP). *J. Geophys. Res. Atmospheres* **119**, 8695–8711 (2014).
- ¹⁸ Yu, X. *et al.* Impacts, effectiveness and regional inequalities of the GeoMIP G1 to G4 solar radiation management scenarios. *Glob. Planet. Change* **129**, 10–22 (2015).
- ¹⁹ Ferraro, A. J. & Griffiths, H. G. Quantifying the temperature-independent effect of stratospheric aerosol geoengineering on global-mean precipitation in a multi-model ensemble. *Environ. Res. Lett.* **11**, 034012 (2016).

References

1. Dell, M., Jones, B. F. & Olken, B. A. Temperature Shocks and Economic Growth: Evidence from the Last Half Century. *Am. Econ. J. Macroecon.* **4**, 66–95 (2012).
2. Burke, M., Hsiang, S. M. & Miguel, E. Global non-linear effect of temperature on economic production. *Nature* **527**, 235–239 (2015).
3. Burke, M., Davis, W. M. & Diffenbaugh, N. S. Large potential reduction in economic damages under UN mitigation targets. *Nature* **557**, 549–553 (2018).
4. Newell, R., Prest, B. & Sexton, S. The GDP Temperature Relationship: Implications for Climate Change Damages. *Resour. Future Work. Pap.* (2018).
5. SREX, I. *Managing the risks of extreme events and disasters to advance climate change adaptation. Intergovernmental panel of climate change, special report.* (Cambridge University Press, Cambridge, UK/New York, 2012).
6. Irvine, P. *et al.* Halving warming with idealized solar geoengineering moderates key climate hazards. *Nat. Clim. Change* **1** (2019). doi:10.1038/s41558-019-0398-8
7. MacMartin, D. G. & Kravitz, B. Dynamic climate emulators for solar geoengineering. *Atmospheric Chem. Phys.* **16**, 15789–15799 (2016).
8. Acemoglu. *Introduction to Modern Economic Growth.* (Princeton University Press, 2009).
9. McLaren, D. P. Whose climate and whose ethics? Conceptions of justice in solar geoengineering modelling. *Energy Res. Soc. Sci.* **44**, 209–221 (2018).
10. MacMartin, D. G., Kravitz, B. & Rasch, P. J. On solar geoengineering and climate uncertainty. *Geophys. Res. Lett.* **42**, 7156–7161 (2015).
11. MacMartin, D. G., Ricke, K. L. & Keith, D. W. Solar geoengineering as part of an overall strategy for meeting the 1.5°C Paris target. *Phil Trans R Soc A* **376**, 20160454 (2018).

12. Tilmes, S. *et al.* The hydrological impact of geoengineering in the Geoengineering Model Intercomparison Project (GeoMIP). *J. Geophys. Res. Atmospheres* **118**, (2013).
13. Kravitz, B. *et al.* An energetic perspective on hydrological cycle changes in the Geoengineering Model Intercomparison Project. *J. Geophys. Res. Atmospheres* **118**, (2013).
14. Kravitz, B. *et al.* A multi-model assessment of regional climate disparities caused by solar geoengineering. *Environ. Res. Lett.* **9**, 074013 (2014).
15. Jones, A. *et al.* The impact of abrupt suspension of solar radiation management (termination effect) in experiment G2 of the Geoengineering Model Intercomparison Project (GeoMIP). *J. Geophys. Res. Atmospheres* **118**, 9743–9752 (2013).
16. Irvine, P. J. *et al.* Key factors governing uncertainty in the response to sunshade geoengineering from a comparison of the GeoMIP ensemble and a perturbed parameter ensemble. *J. Geophys. Res. Atmospheres* **119**, 7946–7962 (2014).
17. Xia, L. *et al.* Solar radiation management impacts on agriculture in China: A case study in the Geoengineering Model Intercomparison Project (GeoMIP). *J. Geophys. Res. Atmospheres* **119**, 8695–8711 (2014).
18. Yu, X. *et al.* Impacts, effectiveness and regional inequalities of the GeoMIP G1 to G4 solar radiation management scenarios. *Glob. Planet. Change* **129**, 10–22 (2015).
19. Ferraro, A. J. & Griffiths, H. G. Quantifying the temperature-independent effect of stratospheric aerosol geoengineering on global-mean precipitation in a multi-model ensemble. *Environ. Res. Lett.* **11**, 034012 (2016).

Reviewers' comments:

Reviewer #1 (Remarks to the Author):

Second review, Harding et al.

I appreciate the effort the authors have made in revising this paper, and in responding to my comments. Some of the presentational aspects are much better, but unfortunately, the substance of the papers has not changed (and admittedly cannot be changed). It all comes down to how defensible the many assumptions and model are.

On the use of robust, I understand that things are robust in the sense of being in the same direction for all models considered, but there are plenty of cases where many models are known to be robustly wrong because they all make similar simplifications, or we don't know whether they are all robustly wrong.

The authors respond to one comment: "this paper is about comparing apples to apples. If an empirical macroeconomic approach is a legitimate one for evaluating the impacts of global warming (as the recent peer-reviewed publications in Nature suggest it is), then there is no scientific reason it should be considered illegitimate for projecting the impacts of solar geoengineering."

Well, first of all, the fact that something has been published in Nature (or even IPCC) does not imply that it is correct and useful. In fact, a large fraction of papers in Nature is later shown to be at least partly wrong. But more importantly, I would be less concerned in comparing a simple impact model across different CO₂ induced warming levels, because at least the physical processes are the same, and even if the impact model is imperfect, the direction of change is probably ok, or at least similarly biased. But here the authors are comparing responses from longwave vs. shortwave radiative forcing. We know these are different, and I argue we have a much poorer understanding of the changes in due to SRM. So it's not just about the impacts models, it's about the inability of GCMs to predict the changes in climate in response to SRM. In that respect this study different.

The issue of extremes was highlighted by both reviewers, but is not really addressed.

The authors defend their work by pointing to a large number of studies resulting from model intercomparisons. While I do appreciate the effort of these model simulations, robust model agreement in this context means virtually nothing. In contrast to CO₂ induced warming, we do not have any observations whatsoever to evaluate the response of those models to SRM. It is entirely plausible that all models are wrong in the same way, because they are all doing the same things in oversimplified ways. The community has been working for 40 years or so on understanding the water cycle in response to CO₂, with a massive effort of collecting data, and we are still struggling even on basic things. Some of the processes involved in the response to SRM are highly idealized in these models, and without any observations to test I just don't have confidence in these. The reality is that this study does not offer a single prediction that could be tested or falsified.

In summary, my view of this paper has not changed much. The title and discussion of uncertainties is much better, but the evidence is the same, and speculative in my view: at the level of GCMs being able to simulate the response to SRM, at the level of parameterizing climate impacts in very simple ways (untested for SRM), and (as the second reviewer noted) in treating inequality in a particular way. As mentioned in my first review, I think this is an interesting topic to look at, and we should prevent publication of such results, but given the many questions marks I recommend publication in a technical journal, along with further analysis the issue of variability and extremes. In the light of the controversial nature and visibility of the topic, I don't think the evidence provided matches the standards I would expect from a Nature journal.

Reviewer #2 (Remarks to the Author):

Accept with minor revisions

The authors have attempted to respond to all the concerns. It is good that they have now mellowed down the message that solar geo-engineering would reduce global income inequality. However, the real issue is not really the economic modelling's applicability to climate impact discussions (these are already published and accepted), it is if these models can be used for solar-geo-engineering related analysis. The authors argue that it could be. However, the possibility of delinking of temperature increase with precipitation due to geo-engineering interventions and what does this mean for climate impacts as well as statistical analysis is critical here. The same models used for climate impacts are reasonable as they invariably assume a link between temperature and precipitation that is reflected in the historical data, but this might not be the case under solar geo-engineering scenarios. However, for the benefit of academic discourse, I would give this paper the benefit of the doubt and recommend for publication with the following suggested revisions:

1. Modify the title to 'Climate econometric models indicate globally governed solar geoengineering may reduce inter-country income inequality'.

First, adding 'globally governed' is critical. Please add a paragraph in the methodology and conclusion highlighting that this study assumes that solar geo-engineering is globally governed with no country unilaterally going ahead and doing this at a large scale. This, though, is fundamentally inconsistent with the SSP3 framing (which is about regional rivalry), and the authors should mention this inconsistency of a globally governed geo-engineering architecture with the SSP3 narrative in their methodology and try to reason that this inconsistency is acceptable for this analysis. They should mention, especially in the conclusion, that on the contrary, an ungoverned series of geo-engineering interventions by resourceful countries for their own interests could lead to a very different outcome that has not been analysed by them.

Second, people read the term 'global' in many ways, and that is not what this study does. It is strictly looking at 'inter-country' GDP and this should be reflected in the title.

2. Please mention somewhere in the abstract explicitly that the impacts of precipitation changes are statistically insignificant, and hence the results are mainly driven by change in average temperatures.

3. Please mention at the appropriate place in the results section if there is any difference in results for any of the SSPs. Reading the results it appears that the same results hold for most but not all SSPs, please clarify. If so, please explain why the result is different. Please mention this difference in the conclusion section as well.

I think with these changes, the paper would be closer to what it actually does and what are the biggest caveats. Let me commend the authors once again for the hard work and very detailed analysis. I am sure this would be a useful contribution to the academic discourse (and might be heavily criticised as well) irrespective of the philosophical differences in the climate community on the issue of geo-engineering.

We thank the referees for their efforts in providing comments about our manuscript, which we respond to below. The referees' comments are in *italics* and our point-by-point responses follow. To facilitate the review, we have appended to the end of this document a track-changes version of our resubmitted manuscript which highlights the revisions.

Reviewer #1 (Remarks to the Author):

Second review, Harding et al.

I appreciate the effort the authors have made in revising this paper, and in responding to my comments. Some of the presentational aspects are much better, but unfortunately, the substance of the papers has not changed (and admittedly cannot be changed). It all comes down to how defensible the many assumptions and model are.

On the use of robust, I understand that things are robust in the sense of being in the same direction for all models considered, but there are plenty of cases where many models are known to be robustly wrong because they all make similar simplifications, or we don't know whether they are all robustly wrong.

Response: We appreciate the reviewer's skepticism (and even share it, to an extent), but this argument could be used to reject 100% of all submitted papers. It's true that sometimes a body of knowledge is proven wrong, but rejecting a paper because of the possibility that the scientific body of knowledge is wrong would stop the scientific process in its tracks.

We address this concern through extensive uncertainty analysis and extensive discussion of the limitations of this approach when applied to both solar geoengineering and GHG-driven climate change. We are agnostic about econometric approach and climate model. Our battery of tests expands beyond all the scientific literature currently published. We believe our approach to estimating macroeconomic impacts of climate is the most exhaustive today in the literature, not only on geoengineering impacts, but as applied to climate change in general.

We discuss extensively in the conclusions the possible places where our results, by relying on the previous literature, may be shown to be wrong. Throughout the paper, we have reframed our results as, "If X is true, then Y follows." But at the moment there is no evidence that the current macroeconomic approaches are indeed wrong. If the referee knows of any such evidence, we would appreciate if they can provide citations.

Reviewer #1: *The authors respond to one comment: "this paper is about comparing apples to apples. If an empirical macroeconomic approach is a legitimate one for evaluating the impacts of global warming (as the recent peer-reviewed publications in Nature suggest it is), then there is no scientific reason it should be considered illegitimate for projecting the impacts of solar geoengineering."*

Well, first of all, the fact that something has been published in Nature (or even IPCC) does not imply that it is correct and useful. In fact, a large fraction of papers in Nature is later shown to be at least partly wrong.

Response: Yes, it is true that sometimes papers in *Nature* turn out to be wrong. However, *Nature* does have a very rigorous review process and it is standard practice in science to build upon the results of previous credibly peer-reviewed work until it has been demonstrated to be incorrect. Nevertheless, we agree with the possibility that some of the literature can be wrong and that is why we have provided an empirical explanation (now moved from the Supplementary Materials to the main text) for why the potential issues with the impacts model capturing the effects of precipitation are equivalent under GHG-driven and solar geoengineering-driven climate change. The papers whose econometric models we apply to project economic outcomes¹ have large Supplementary Texts associated with them that conduct extensive robustness checks (authors' phrasing, not ours). Again, it would help us respond to the reviewer's concerns better if they could provide citations to illustrate their criticisms.

Reviewer #1: *But more importantly, I would be less concerned in comparing a simple impact model across different CO2 induced warming levels, because at least the physical processes are the same, and even if the impact model is imperfect, the direction of change is probably ok, or at least similarly biased. But here the authors are comparing responses from longwave vs. shortwave radiative forcing. We know these are different, and I argue we have a much poorer understanding of the changes in due to SRM. So it's not just about the impacts models, it's about the inability of GCMs to predict the changes in climate in response to SRM. In that respect this study different.*

Response: First, it's important to note that our analysis does not look at outcomes under a single simple impact model, but rather a number of impact models. Economic impact models of the type we apply differ quite significantly from GCMs, in principle. Whereas with GCMs there is a level of agreement among scientists and modeling groups about the fundamental physics, the ten impacts models that we apply represent fundamentally different views of how the climate impacts the economy. As we show in the manuscript, this results in a very diverse set of absolute economic outcomes and yet, despite this, the inequality results hold.

Next, our understanding of the physical climate science literature differs from the referee's reading, and so we have provided citations to previous work to support our conclusions, both within the manuscript and below. In our view it is not accurate that research on climate change

¹ Burke, M., Hsiang, S. M., & Miguel, E. (2015). Global non-linear effect of temperature on economic production. *Nature*, 527(7577), 235; Dell, M., Jones, B. F., & Olken, B. A. (2012). Temperature shocks and economic growth: Evidence from the last half century. *American Economic Journal: Macroeconomics*, 4(3), 66-95; Newell, R. G., Prest, B. C., & Sexton, S. (2018). The GDP-Temperature Relationship: Implications for Climate Change Damages. RFF Working Paper. Available at: <http://www.rff.org/research/publications/gdp-temperature-relationship-implicationsclimate-change-damages>.

has focused predominantly on longwave and not shortwave forcings. The presence and growth of both long- and shortwave forcings over the past century is one of the central challenges to constraining climate sensitivity and improving climate models, precisely because it has been difficult to disentangle the effects of either in the presence of both.

For an example of contemporary analysis related to this topic see this paper by Marvel et al, published this spring: <https://www.nature.com/articles/s41586-019-1149-8> . They demonstrate explicitly the importance of considering both long- and shortwave (aerosol) anthropogenic forcings to understand the anthropogenic influence on hydrological variables. Observations of the hydrological response to volcanic eruptions (the analog forcing that is closest in model implementation to solar geoengineering) have been applied to improving GCMs dating back to just after Pinatubo (see, e.g., Hansen et al (1996): https://link.springer.com/chapter/10.1007/978-3-642-61173-5_20) and precipitation response to volcanic eruptions in CMIP models has also been extensively interrogated (see e.g., Barnes et al (2016): <https://journals.ametsoc.org/doi/full/10.1175/JCLI-D-15-0658.1>). In addition, a great deal of work has been done on the regional responses of the hydrological cycle to shortwave forcings from anthropogenic aerosols from fossil fuels and other pollution (see, e.g., Westervelt et al (2018) for a compelling recent treatment of this topic: <https://www.atmos-chem-phys.net/18/12461/2018/>). These are just some examples of high-quality publications in top journals that do not support the referee's contention that climate science has not examined responses to shortwave forcing.

Uncertainty about the role of longwave versus shortwave forcings in driving (in particular, regional) hydrological responses in the presence of substantial natural variability is a fundamental problem in projecting impacts of climate change either with or without solar geoengineering. We use climate models to make the best projections we can and explore the uncertainty across an ensemble, as is the standard practice in contemporary climate science. To emphasize that all of the results are generated from uncertain models, we have added the following sentence to the final paragraph of the main text:

There are substantial uncertainties associated with the models applied in this study, but the reduction of inter-country inequality is consistent across socioeconomic scenario, climate model and economic model combinations.

We have also added the following sentence to the penultimate paragraph of the main text:

These conclusions are dependent on the historically-trained climate-econometric models being valid in predicting future impacts of geoengineering, but if these models are not valid for geoengineering, our results suggest we should also expect them to also be invalid for GHG-driven climate change.

Reviewer #1: *The issue of extremes was highlighted by both reviewers, but is not really addressed.*

Response: An article in *Nature Climate Change* was published in March that is entirely focused on analysis of extremes in solar geoengineering simulations. In our previous revisions and response to comments we highlighted and cited this recent work. We now call further attention to its results with an additional sentence:

Factors such as climate variability and extremes are only captured by this model to the extent that they are related to the climate indicators used in these models. We cannot partition these effects from the aggregate effects using the empirical impacts estimation models we apply, and as such, considering the impact of these this is outside the scope of our analysis. **However, recent work using a high-resolution forecast-oriented model found that the type of solar geoengineering simulated in the GeoMIP simulation ensemble (which we apply as well) mediates precipitation extremes over 99.6% of grid cells and reduces tropical cyclone intensity, not just mean climate response, supporting the assumption that there is a strong relationship between reduction of mean anomalies and mitigation of extremes¹⁷.**

The reason we do not focus on extremes is because empirical econometric impact models have generally identified the impact of climate on the economy using annual mean temperature and precipitation. We cannot apply these state-of-the-art models using extremes and the purpose of this paper is not to identify new drivers of economic impacts but rather apply these same models to a novel set of climate projections.

Reviewer #1: *The authors defend their work by pointing to a large number of studies resulting from model intercomparisons. While I do appreciate the effort of these model simulations, robust model agreement in this context means virtually nothing. In contrast to CO₂ induced warming, we do not have any observations whatsoever to evaluate the response of those models to SRM. It is entirely plausible that all models are wrong in the same way, because they are all doing the same things in oversimplified ways.*

Response: This comment is difficult to address. If agreement among suites of both state-of-the-art physical and economic models means “virtually nothing,” where can we start?

We believe we may have a fundamental philosophical difference with the reviewer, who appears to set a different standard for geoengineering modeling studies than for other climate modeling studies. Unfortunately, this double standard is antithetical to our work because we aim to begin standardizing quantitative impacts assessment of geoengineering and CO₂-driven warming. We respect the reviewer’s different opinion in this matter, but we do not share it.

Reviewer #1: *The community has been working for 40 years or so on understanding the water cycle in response to CO₂, with a massive effort of collecting data, and we are still struggling even on basic things. Some of the processes involved in the response to SRM are highly idealized in these models, and without any observations to test I just don't have confidence in these.*

Response: Again, see our response above about the literature on the response of the hydrological cycle to short- and longwave forcing. We differ on whether the community has truly focused solely on CO₂, as opposed to forcings more generally. We would appreciate if the referee could provide citations to guide us forward in interpreting their claims. Following the reviewer's concern to its logical conclusion, we should not publish papers on geoengineering impacts until we've actually done it in the real world, but one could make similar claims about projections of climate change. Many things in the models (for example, permafrost thaw) are highly idealized, without any observations, yet papers are published with appropriate caveats and uncertainty analysis.

Reviewer #1: *The reality is that this study does not offer a single prediction that could be tested or falsified.*

Response: We certainly agree, but no climate modeling study with projections out to the late 21st century offers a prediction that can be tested or falsified. In the present and recent past, an estimated 20-60% of the positive longwave forcing from greenhouse gases are masked by shortwave forcings. Despite these large uncertainties, we use our earth system models to project scenarios of future change with both long- and shortwave forcings outside the empirically falsifiable set of projections because of the societal importance of producing the best projections possible given the state of the science.

Reviewer #1: *In summary, my view of this paper has not changed much. The title and discussion of uncertainties is much better, but the evidence is the same, and speculative in my view: at the level of GCMs being able to simulate the response to SRM, at the level of parameterizing climate impacts in very simple ways (untested for SRM),*

Response: We agree with the referee that results produced by empirical macroeconomic impact models are speculative, which is why we have written this manuscript using solar geoengineering to illustrate some unintuitive results if you apply the models and why we devote such a large portion of our text to exploring the implication of model specification uncertainty on the outcomes of interest -- in simulations both with and without solar geoengineering.

We have used all of the climate models in the GeoMIP simulation ensemble. We have applied ten econometric model specifications from multiple leading economic modeling groups. To our knowledge, these are the most comprehensive set of climate outcomes in this topic area. In that sense, the evidence is no more speculative than any other climate model projection, but

captures scientific uncertainties in a way that is consistent with the practices of contemporary climate science.

As we write in our conclusions:

If our application and results induce skepticism, this may indicate that the empirical macroeconomic impacts assessment approach is inappropriate to apply in projecting future climate damages in general, whether solar geoengineering is a component of that future change or not. If this modeling approach accurately identifies the climate-economy relationship independent of the driving cause of climate variation, then empirical macroeconomic impacts models indicate that solar geoengineering is a tool that could be applied to grow the global economy and reduce global income inequality. There is no scientific reason that this empirical modeling approach and resulting climate change impact projections would be appropriate to apply in one instance and not the other.

We have applied all of the econometric approaches currently discussed in the literature (as well as some additional cases to push the boundaries of this literature). Our parameterizations are state-of-the-art and goes beyond any one particular approach. We agree, the methodology is untested for solar geoengineering and that is why we are writing this paper.

In order to highlight this further, and increase the ease with which readers can interpret the econometric model uncertainty analysis, we have generated an additional figure (Figure 4) for the revised manuscript:

Figure 4. Percentage of countries with a relative loss compared to no climate change versus country-level Gini Coefficients in 2099. Values represent median projections for SSP3 for RCP8.5

(red), geoengineering-stabilized RCP8.5 (green) and geoengineering-mirrored RCP8.5 (blue) simulations. Numbers represent models specified as follows: (1) Estimates a pooled growth model with quadratic temperature and precipitation, year fixed effects, and a quadratic country time trend. (2) Estimates a growth model with quadratic temperature and precipitation and lags up to 5 years, year fixed effects, and a quadratic country time trend. (3) Estimates a growth model with quadratic temperature and precipitation for rich and poor countries separately, year fixed effects, and a quadratic country time trend. (4) Estimates a growth model with quadratic temperature and precipitation for rich and poor countries separately lagged up to 5 years, year fixed effects, and a quadratic country time trend. (5) Estimates a growth with linear temperature separately for rich and poor countries, region-year fixed effects, and no country time trend. (6) Estimates a pooled growth model with quadratic temperature, region-year fixed effects, and no country time trend. (7) Estimates a pooled growth model with quadratic temperature and precipitation, region-year fixed effects, and no country time trend. (8) Estimates a pooled growth model with quadratic temperature and precipitation lagged up to 5 years, region-year fixed effects, and no country time trend. (9) Estimates a pooled levels model with quadratic temperature and precipitation, region-year fixed effects, and a quadratic country time trend. (10) Estimates a pooled levels model with quadratic temperature and precipitation, year fixed effects, and a quadratic country time trend. (11) Estimates a pooled levels model with quadratic temperature and precipitation, region-year fixed effects, and no country time trend.

We have inserted the accompanying text:

The effects of each scenario on country-level economic growth and inter-country inequality varies by economic impact model specification (Figure 4). While the results above suggest that more countries are economically better off under scenarios with geoengineering, the percentage of countries absolutely or relatively poorer varies across economic impact models (see also Supplementary Table S4 and S5). As shown by Figure S3 in the Supplementary Materials, the identity of countries that experience economic losses and the magnitude of their absolute or relative losses also varies across models. Unlike projections of global economic growth over the next century, projections of global income inequality are qualitatively consistent across models, suggesting that using solar geoengineering to negate or reverse climate change can reduce global income inequality. **Figure 4 displays the percentage of countries that gain relative to no climate change and the Gini coefficients for country GDP/capita in 2099 for the different econometric models and illustrative climate scenarios under SSP3.** Gini coefficients are a widely used measure of inequality, related to the curvature of the Lorenz curves in Figure 3, where a lower Gini coefficient indicates lower inequality. **Despite significantly disparate models of how climate impacts economic growth, several consistent trends emerge. RCP8.5 consistently increases inter-country inequality and the percentage of countries with poor economic growth, whereas the geo-mirrored scenario consistently decreases inequality. For all impact models, the Gini coefficient decreases with the use of solar geoengineering. The coefficient is the lowest for the Geoengineering Mirrored RCP 8.5 scenario. (Figure S7 in the Supplementary Materials displays the Lorenz curves across different model specifications.) Under all but one economic impacts model, the Geoengineering Mirrored RCP 8.5 scenario decreases the percentage of countries with a GDP loss relative to RCP8.5, and under that particular model**

(an income-dependent growth model with no country time trends), geoengineering has a particularly large effect on reducing inequality.

Reviewer #1: *and (as the second reviewer noted) in treating inequality in a particular way.*

Response: We have addressed Reviewer #2's comments on our treatment of inequality and explicitly discuss why we treat inequality in this way. Reviewer #2 recommends publication.

Reviewer #1: *As mentioned in my first review, I think this is an interesting topic to look at, and we should prevent publication of such results, but given the many questions marks I recommend publication in a technical journal, along with further analysis the issue of variability and extremes. In the light of the controversial nature and visibility of the topic, I don't think the evidence provided matches the standards I would expect from a Nature journal.*

Response: We appreciate the Reviewer's candor about the fact that their recommendation for rejection is based partially on the controversial nature and visibility of the project, but we disagree with the reasoning that this is a legitimate reason to reject publication of geoengineering research. We agree that this is an important and interesting topic. Advancing our understanding of the issue requires that we use literature in different sub-fields and carefully apply it to a new topic. We believe the most interesting advances come from cross-disciplinary approaches to conventional questions. We believe our paper fits that description and it is because of the cutting edge nature of our paper that we believe *Nature Communications* is a perfect fit.

Reviewer #2 (Remarks to the Author):

Accept with minor revisions

The authors have attempted to respond to all the concerns. It is good that they have now mellowed down the message that solar geo-engineering would reduce global income inequality. However, the real issue is not really the economic modelling's applicability to climate impact discussions (these are already published and accepted), it is if these models can be used for solar-geo-engineering related analysis. The authors argue that it could be. However, the possibility of delinking of temperature increase with precipitation due to geo-engineering interventions and what does this mean for climate impacts as well as statistical analysis is critical here. The same models used for climate impacts are reasonable as they invariably

assume a link between temperature and precipitation that is reflected in the historical data, but this might not be the case under solar geo-engineering scenarios.

Response: We thank the referee for their encouraging comments. We have added substantially more material to the Main Text to clarify the implications of the dual issues of: (1) non-significance of precipitation to economic impact, and (2) changing relationships between temperature and precipitation under future climate change scenarios. This includes moving what was formerly Supplementary Figure S8 to include it now as Figure 5. In particular, we have tried to illustrate explicitly that the decoupling of temperature and precipitation under climate change is not more significant with solar geoengineering than greenhouse gas-driven warming. Rather, through the process of reconciling these impacts models with solar geoengineering science, we identify a problem that has been present in the climate econometric literature all along:

The impacts solar geoengineering may have on global and regional hydrological cycle has been a focus of considerable study and concern over the past decade³²⁻³⁵. This study and others have found limited effects of precipitation on economic growth^{11,36,37} (Supplementary Figure S10), meaning the results are mainly driven by change in average temperatures. Both greenhouse gas-driven warming and solar geoengineering are expected to decouple the historical regional relationships between temperature and precipitation in a way that is not necessarily well-accommodated by empirical impacts models. With the decoupling of both precipitation and other impact-relevant variables under solar geoengineering, historical relationships measured with the usual climate indicators used in climate change analyses may no longer be appropriate. Importantly, this decoupling is not unique to solar geoengineering. **While historically, annual precipitation and temperature are negatively correlated most areas over land, the same is not necessarily true in future change scenarios either with or without solar geoengineering (Figure 5).** Lack of cross-sectional variation in correlations could prove problematic when projections are then made using a model that includes country fixed effects^{38,39} in which the value of a base climate state are aggregated with the value of non-physical properties such as economic and political institutions. Under projections for both GHG-driven climate change and solar geoengineering, the relationship between precipitation and temperature is distinctly different than historical interannual variability. The sign of the relationship between precipitation and temperature changes for nearly half of all countries in the analysis for both projections relative to the historical relationship (76 countries for RCP8.5 and 73 for solar geoengineering out of 165 countries). This suggests that extrapolation based solely on historical observation may be inaccurate in its representation of future climate-economy impacts due to a sharply changing relationship between temperature and precipitation. **This problem is not unique to solar geoengineering, but is rather of equal consequence for projections of both the impacts of GHG-driven change and solar geoengineering.**

Fig 5. Historical and projected relationship between surface temperature and annual precipitation. (a) Historical correlation between temperature and precipitation. Change in precipitation relative to change in temperature projected by (b) a CMIP5 ensemble for RCP8.5 and by (c) a GeoMIP ensemble for solar geoengineering to reduce the global the mean temperature by an equal amount as the warming under RCP8.5.

Reviewer #2: *However, for the benefit of academic discourse, I would give this paper the benefit of the doubt and recommend for publication with the following suggested revisions:*

1. Modify the title to 'Climate econometric models indicate globally governed solar geoengineering may reduce inter-country income inequality'.

First, adding 'globally governed' is critical. Please add a paragraph in the methodology and conclusion highlighting that this study assumes that solar geo-engineering is globally governed with no country unilaterally going ahead and doing this at a large scale. This, though, is fundamentally inconsistent with the SSP3 framing (which is about regional rivalry), and the authors should mention this inconsistency of a globally governed geo-engineering architecture with the SSP3 narrative in their methodology and try to reason that this inconsistency is acceptable for this analysis. They should mention, especially in the conclusion, that on the contrary, an ungoverned series of geo-engineering interventions by resourceful countries for their own interests could lead to a very different outcome that has not been analysed by them. Second, people read the term 'global' in many ways, and that is not what this study does. It is strictly looking at 'inter-country' GDP and this should be reflected in the title.

Response: We agree with the referee that “inter-country income inequality” is a better way to characterize our analysis and have made this change.

We have not made the “globally governed” addition, however, because we are not sure that it is correct, and believe making that claim is beyond the scope of this paper. Historically, global governance structures uphold existing power dynamics; that is, the countries with the most power build institutions which will help them preserve that power. Under the assumption of a globally governed process, therefore, the kind of idealized “over cooling” scenario we model would probably be less likely than a more moderate cooling scenario with less income inequality reduction (such as the geo-stabilized scenario). It could be more likely to see global cooling implemented under a geopolitical scenario wherein a coalition of equatorial countries, perhaps including one nuclear power, act without the full consent of all nations. In any case, this is mostly speculative. As we state in our paper, these are “highly idealized” illustrative scenarios. We have neither the expertise nor sufficient space to get into substantive discussion of governance scenarios in this paper, beyond pointing out that it is an important challenge to address as solar geoengineering research progresses. To this end, we have added the following sentences to the final paragraph of the main text highlighting the difficulty of governance of SRM and emphasizing that the scenarios are highly idealized:

We have presented results based on idealized scenarios that are unlikely to be politically or legally feasible. However, the strategic incentives implied by the results highlight the need for further work on the global governance of solar geoengineering.

Reviewer #2: 2. Please mention somewhere in the abstract explicitly that the impacts of precipitation changes are statistically insignificant, and hence the results are mainly driven by change in average temperatures.

Response: Due to the abstract word limit, we’ve addressed this in the abstract just by adding that the results are “temperature-driven”. However, as described above in our response to Reviewer #2’s first comment, we have expanded the main text to delve more deeply into the twin challenges of non-significance of precipitation and shifting correlations between temperature and precipitation in the climate change projections.

Reviewer #2: 3. Please mention at the appropriate place in the results section if there is any difference in results for any of the SSPs, Reading the results it appears that the same results hold for most but not all SSPs, please clarify. If so, please explain why is the result different. Please mention this difference in the conclusion section as well.

Response: Yes, the results are qualitatively consistent (in their effect of inequality) between all five SSPs. We have added the following sentence to the results sections to clarify this point:

Likewise, this result is consistent among all SSPs.

Reviewer #2: *I think with these changes, the paper would be closer to what it actually does and what are the biggest caveats. Let me commend the authors once again for the hard work and very detailed analysis. I am sure this would be a useful contribution to the academic discourse (and might be heavily criticised as well) irrespective of the philosophical differences in the climate community on the issue of geo-engineering.*

Response: We thank the referee for their encouraging comments.

Reviewers' comments:

Reviewer #2 (Remarks to the Author):

The author's have done good work in terms of the responding to the reviewer's comments. My comments in this round are focused on author's response to my comments. The authors have unfortunately not responded to the most important comment of mine- that of mentioning in the title and the text that the analysis implicitly assumes that geoengineering is globally governed. The following are my detailed comments related to this:

1. Methodologically, it is very clear that authors have assumed ideal scenarios. That is, geoengineering interventions don't end up playing havoc like significantly altering monsoon patterns in the tropics because the interventions have not been planned in the 'right' way (a paper's link is given in point no. 3). The authors also mention that the scenarios are idealised scenarios. All this clearly implies that the authors are referring to globally governed geo-engineering scenarios, and nothing else has been explored by them.
2. This paper is not about the science of geoengineering, but the policy aspects of it. The most critical policy debate happening in the geoengineering related forums is related to governance of this approach, many papers on this can be found in the current literature. The authors might not be experts on geoengineering governance, but they can not simply ignore it because they are not experts. This, or any other, policy related paper is adding to the existing literature on policies related to geo-engineering. The authors have to provide a link to contextualize their effort to the existing debate, and that link is governance. The authors might not have explicitly explored it, but an ideal globally governed geo-engineering paradigm is implicitly there in all their scenarios, and it is important to spell this out clearly.
3. Based on the above points, it appears to me that I was probably not sufficiently clear, or maybe I didn't emphasise it enough in my earlier comments, though I think I mentioned this clearly. Maybe that is why authors have assumed that this was a minor comment. I want to mention clearly and emphatically now, that in any policy oriented paper like this, one can't simply ignore the issue of how geoengineering interventions are transitioned from labs to reality. The worst think that could happen as an outcome of this paper being published is that one or two of the rich and powerful countries of the world use the results of this paper as an argument to unilaterally go ahead with large scale geoengineering interventions, as the paper argues that it has tested a huge bunch of scenarios and uncertainties and that it robustly concludes that geo-engineering leads to reduction in inter-country inequality. This conclusion, if some parties try to generalise it, can not be further away from the truth, as what the authors have tested is idealised scenarios, which in other words definitely means 'only' globally governed geoengineering interventions. I would request the authors to reflect on this. It is our responsibility as researchers to be aware of the larger debate, especially on the issue as controversial as geo-engineering, and ensure that our research is placed within that context, and is not read out of this context. An intervention and its impact like that described in this paper by Nalam et al. <https://link.springer.com/article/10.1007/s00382-017-3810-y> is undesirable and not what the authors are alluding to in their results. The authors should be cognisant of this.
4. The authors have assumed that my comments imply that the paper is accepted. I want to clarify that in my view, the paper should be accepted only if it clearly mentions in the title as well as in the relevant sections throughout the paper that the insights are applicable to only a globally governed geoengineering, which is akin to the idealised scenarios, rather than unilateral moves by any country. In this kind of paper, it is critical to be clear and upfront about this, rather than hope that people will read between the lines, because some stakeholders could also ignore what ever if between the lines even if they understand it, only for the sake of supporting their argument.
5. To summarise, as I mentioned in my earlier comments, the paper is going to be controversial,

but authors have done a good academic work, and it will definitely contribute to the debate. However, it is the responsibility of the authors to ensure that the key message should not be read and interpreted out of context. For that, including the globally governed angle in the title, abstract, conclusion, and other relevant parts of the paper is critical. This journal is a high impact journal, and I expect this paper, once published, will end up becoming a highly cited paper. Thus, the responsibility that the results are not misrepresented by any party is with the authors. I would like to accept the paper only if it includes the suggested change.

Reviewer

The author's have done good work in terms of the responding to the reviewer's comments. My comments in this round are focused on author's response to my comments. The authors have unfortunately not responded to the most important comment of mine- that of mentioning in the title and the text that the analysis implicitly assumes that geoengineering is globally governed. The following are my detailed comments related to this:

1. Methodologically, it is very clear that authors have assumed ideal scenarios. That is, geoengineering interventions don't end up playing havoc like significantly altering monsoon patterns in the tropics because the interventions have not been planned in the 'right' way (a paper's link is given in point no. 3). The authors also mention that the scenarios are idealised scenarios. All this clearly implies that the authors are referring to globally governed geo-engineering scenarios, and nothing else has been explored by them.

Thank you to the reviewer for raising our attention to this important implicit assumption as a result of our modeling decisions. We agree that there are implicit assumptions made especially for the deployment strategy of solar geoengineering that should be acknowledged in the paper. While we disagree that the scenarios are “ideal” (a term which we previously misused in the paper and have adjusted accordingly), now that we better understand the reviewer’s motivations for including language about global governance, let us explain why we believe it would be misleading and incorrect, from an international relations theory perspective, to use the term “globally governed” to describe our scenarios.¹

Some of the biggest losers under conditions of economic convergence are rich countries. This is because, by definition, under conditions of convergence rich countries get poorer as poor countries get richer. Foundational theories of international relations (IR) suggest that because rich countries are also the most powerful ones, they have historically controlled global governance, and will continue to do so as long as they are comparatively richer/more powerful than other countries. IR theories further suggest that rich/powerful countries are rational actors, and as such make decisions based in their own self-interest, which is largely determined by economic indicators, such as maximizing GDP. Therefore, it is very unlikely that rich countries who control global governance would steer it towards the *RCP8.5-mirrored* scenario because this scenario, according to our findings, makes them poorer (in both absolute and relative terms). As such, the *RCP8.5-mirrored* scenario is unlikely to emerge under conditions of global governance. Therefore, when we said our results are based on an “idealized” scenario, we meant the opposite from what the reviewer is suggesting with respect to global governance. That is, the scenario is idealized because it bucks what is politically likely in future, given what we know from IR theory

¹ Many thanks to international relations scholar, Sikina Jinnah, for her help here in crafting a coherent explanation of “global governance,” based on IR theory and her essential feedback while revising our manuscript. Thanks as well, to David Victor, for his input and perspective on our manuscript and its relation to global governance debates around geoengineering.

about how states are likely to behave. So, rather than our model operating for “only globally governed interventions” as the reviewer suggests, quite the opposite is true. The mirroring scenario is only plausible in the absence of global governance.

That said, we understand, and indeed share, the reviewer’s deeper concern about the policy implications of these findings. We too would not want our findings about inter-country equity used to justify unilateral deployment by some sub-set of the countries that both has the technical capacity deploy and would benefit most from convergence, if such a subset ever exists in future. We have therefore revised the paper to reword any ambiguous language that might be taken to justify such an action. In addition, in line with standard practice in climate modeling papers, we have included reference to these policy implications and the bounds of our findings, not in the title and abstract, which are meant to report the core scientific/modeling aims of the paper, but rather with additional paragraphs in the conclusions where policy implications of findings are typically discussed. The details of these revisions are as follows:

1. With regard to our scenarios being “ideal”, we argue that this would imply use of optimization which we do not conduct. We have previously misused the word “idealized” in our manuscript, when what we meant was “stylized,” or “depicted or treated in a mannered and nonrealistic style.” We model our scenarios on one that is routinely applied by climate modelers, the RCP8.5 scenario. The two novel illustrative geoengineering scenarios whose effect and impacts we simulate only make sense in the context of the RCP8.5 scenario. We now describe these scenarios as “stylized” in the abstract and throughout the manuscript (seven times, including in the introduction, methods, results, discussion and conclusions). For example:
 - a. In the introduction, we write, “To comparatively evaluate the impacts solar geoengineering with climate change impacts, we construct stylized climate scenarios from climate change and solar geoengineering projections widely used in impacts assessment.”
 - b. In the results section we write, “These stylized scenarios were designed to illustrate the comparison of solar geoengineering with RCP 8.5, a climate change scenario commonly utilized in climate change impact assessment.”
 - c. In the final paragraph we write: “We have presented results based on stylized scenarios that are unlikely to be politically or legally feasible.”
2. We have added two paragraphs and five references to the Conclusions section of the manuscript. These begin (around line 295), “Our findings indicate a potentially large global economic gain from solar geoengineering, if implemented. This does not necessarily indicate that a globally governed deployment strategy would resemble our stylized scenarios. Heterogeneous impacts suggest that the scenario with greatest global economic growth may not be politically feasible under a globally governed system....”

3. We have added a new final sentence to the Main Text of the manuscript in order to end the paper on a note emphasizing governance concerns: “Following the extensive body of literature on solar geoengineering governance, our findings underscore that a robust system of global governance will be necessary to ensure that any future decisions about solar geoengineering deployment are made for collective benefit.”

In conclusion, the reviewer’s concerns about the paper potentially being misinterpreted are entirely fair. We share this concern and do not want our manuscript to be misinterpreted either. While we agree that it is important to address governance in a paper about solar geoengineering, we disagree that our modeling assumptions imply or refer to globally governed scenarios. The deployment strategy may matter and the role of deployment and its dependence on governance structure is an important area of future work but is beyond the scope of this analysis. We more completely address the reviewers concern about governance structure in the next point.

2. This paper is not about the science of geoengineering, but the policy aspects of it. The most critical policy debate happening in the geoengineering related forums is related to governance of this approach, many papers on this can be found in the current literature. The authors might not be experts on geoengineering governance, but they can not simply ignore it because they are not experts. This, or any other, policy related paper is adding to the existing literature on policies related to geo-engineering. The authors have to provide a link to contextualize their effort to the existing debate, and that link is governance. The authors might not have explicitly explored it, but an ideal globally governed geo-engineering paradigm is implicitly there in all their scenarios, and it is important to spell this out clearly.

We agree with the reviewer that the governance of solar geoengineering is an extremely important and active area of policy debate that should be addressed in a paper about solar geoengineering. We respectfully disagree that ours is not a scientific paper. While our work does not primarily focus on physical science, it adheres to social scientific methods for climate change impacts analysis. Thus, while it is policy-relevant, it is not policy-focused. For this reason, we had previously only briefly touched on governance in the conclusions, emphasizing the need for future work accommodating the findings of our analysis. However, in order to address the reviewer’s concerns we have revised the manuscript to provide a more complete discussion of governance, in order to put our analysis fully in context of this body of scholarship.

While we argue that the work in this paper does not implicitly assume a “globally governed geoengineering paradigm,” it does implicitly make assumptions about deployment. To be able to claim that this deployment would be the result of a globally governed strategy would require explicit analysis and evaluation of such an architecture. This holds for any governance structure ranging all the way to uncoordinated unilateral deployment. In this paper, we do not do any such analysis because it is outside the

purpose of this paper as well as outside of the coauthors' areas of disciplinary expertise and training. Our methodology and results do have implications for evaluation of such governance structures, but again this is outside the scope of our analysis. These are important topics for future research, but the goal of this paper is simply to provide an initial comparative analysis of the economic impacts of solar geoengineering using this state-of-the-art empirical methodology, something which has never been done before.

To address these points and make them clear to the reader we have added the following paragraph to the Conclusions:

For purposes of this analysis we generated stylized geoengineering scenarios based on those that have been widely used by climate modelers because our interest was to explore how extreme geoengineering might affect economic growth and inequality. Among the many additional important questions that are beyond the scope of the analysis is how the exact kinds of geoengineering interventions might affect these same outcomes. Already in the broader literature, some scholars have imagined ideal global geoengineering schemes while others see geoengineering emerging in more haphazard ways—initially with actions by governments that may act unilaterally and then, later, with a wider group that sees systemic responses as better than uncoordinated unilateral actions³⁹⁻⁴². Understanding whether and how these different kinds of deployment scenarios is an important topic for future research⁴³.

3. Based on the above points, it appears to me that I was probably not sufficiently clear, or maybe I didn't emphasise it enough in my earlier comments, though I think I mentioned this clearly. Maybe that is why authors have assumed that this was a minor comment. I want to mention is clearly and emphatically now, that in any policy oriented paper like this, one can't simply ignore the issue of how geoengineering interventions are transitioned from labs to reality. The worst think that could happen as an outcome of this paper being published is that one or two of the rich and powerful countries of the world use the results of this paper as an argument to unilaterally go ahead with large scale geoengineering interventions, as the paper argues that it has tested a huge bunch of scenarios and uncertainties and that it robustly concludes that geo-engineering leads to reduction in inter-country inequality. This conclusion, if some parties try to generalise it, can not be further away from the truth, as what the authors have tested is idealised scenarios, which in other words definitely means 'only' globally governed geoengineering interventions. I would request the authors to reflect on this. It is our responsibility as researchers to be aware of the larger debate, especially on the issue as controversial as geo-engineering, and ensure that our research is placed within that context, and is not read out of this context. An intervention and its impact like that described in this paper by Nalam et al. <https://link.springer.com/article/10.1007/s00382-017-3810-y> is undesirable and not what the authors are alluding to in their results. The authors should be cognisant of this.

We apologize for overlooking the importance of the reviewer's concerns in our previous revisions. We agree that we have failed to explain critical implicit assumptions that seemed clear to us as the authors but could be misinterpreted by readers leading to our

results being misused. To mitigate this concern, we have carefully revised the manuscript and reworded important areas of the paper to improve clarity. As we address in a more complete manner in response to the reviewer's specific comments above, we have revised the manuscript to explain implicit assumptions stemming from our modeling choices and have subsequently added the important caveat that our results may be dependent on deployment mode. We also expand the discussion and conclusions to recognize the importance of governance structures for solar geoengineering.

4. The authors have assumed that my comments imply that the paper is accepted. I want to clarify that in my view, the paper should be accepted only if it clearly mentions in the title as well as in the relevant sections throughout the paper that the insights are applicable to only a globally governed geoengineering, which is akin to the idealised scenarios, rather than unilateral moves by any country. In this kind of paper, it is critical to be clear and upfront about this, rather than hope that people will read between the lines, because some stakeholders could also ignore what ever if between the lines even if they understand it, only for the sake of supporting their argument.

As discussed in our response to the reviewer's comments above, our paper makes assumptions about deployment mode, however these do not stem from any clear assumptions about governance structure. We have clarified in the revised manuscript that it is not our intention to imply that the stylized scenarios and models we consider would be the result of globally governed geoengineering. In fact we suspect it will not, nor that our simulated outcomes would be the result of strategic choice of a unilateral move by uncoordinated actors. In the paper we avoid making any such claims and instead emphasize that we consider these stylized scenarios for illustrative purposes only. We state in our revisions that the governance structure may influence or be influenced by our findings and that this is an important area of future research.

As with any climate change impacts paper, this paper has important policy implications which need to be thoroughly discussed. Following the structure of scientific literature we limit these discussions to the conclusion of the paper and focus the title, abstract and results sections on our own scientific findings and aims. Regarding the assumptions we make, both explicitly and implicitly, we address them when introducing each element of our methodology and caveat the results in our discussion. However, we maintain that, in accordance with the standard in the literature, the title is limited to an explanation of the essential modeling aims and findings of the paper and thus refrain from including "idealized" or "globally governed" which would be misrepresentative of the analysis and intent, as well as contrary to a large body of scholarship on international relations theory.

5. To summarise, as I mentioned in my earlier comments, the paper is going to be controversial, but authors have done a good academic work, and it will definitely contribute to the debate. However, it is the responsibility of the authors to ensure that the key message should not be read and interpreted

out of context. For that, including the globally governed angle in the title, abstract, conclusion, and other relevant parts of the paper is critical. This journal is a high impact journal, and I expect this paper, once published, will end up becoming a highly cited paper. Thus, the responsibility that the results are not misrepresented by any party is with the authors. I would like to accept the paper only if it includes the suggested change.

We share their concern for misrepresentation of the findings in this paper and we appreciate the reviewer's persistence in ensuring we mitigate this possibility. As discussed in response to concerns raised by the reviewer above, we have revised and expanded the manuscript adjusting wording and adding discussion where necessary to improve clarity and provide the necessary caveats regarding our analysis as well as acknowledge the important policy implications of our results. Following the common structure of climate modeling and impacts papers, these discussions are limited in the title and abstract, and constrained primarily to the conclusions of the paper where they are typically addressed.

REVIEWERS' COMMENTS:

Reviewer #2 (Remarks to the Author):

I have read through the revised paper, as well as the authors' rebuttal. I am satisfied with the response and the revisions. The authors have now adequately highlighted the global governance debate and how it could be linked to their analysis in their opinion. Also the limitations in terms of the results being invariant to impacts on precipitation are clear.

Philosophical disagreements on the use of geoengineering as well on the robustness of climate econometric models aside, I think this is a very good work and adds on to the debate. I expect the results to be widely debated in the geo-engineering community. The authors should also be ready for the results be outrightly rejected by many in the climate community on the grounds that we don't know enough about the climate system and its response to such interventions, to say anything about economic impacts of solar geoengineering. I also understand that one paper will not and can not change the nature of the debate, it will only provide information to carry forward research on both sides of the debate. I congratulate the authors on this interesting work.